# Digital Twin Technology in Data Center Simulations: Evaluating the Feasibility of a Former Mine Site

Hajime Ikeda [1,*], Nur Ellisha Binti Mokhtar [1], Brian Bino Sinaice [1], Muhammad Ahsan Mahboob [2], Hisatoshi Toriya [1], Tsuyoshi Adachi [1] and Youhei Kawamura [3]

[1] Graduate School of International Resources, Akita University, 1-1, Tegatagakuen machi, Akita-City 0108502, Akita, Japan; bsinaice@rocketmail.com (B.B.S.); toriya@gipc.akita-u.ac.jp (H.T.); adachi.t@gipc.akita-u.ac.jp (T.A.)

[2] Sibanye-Stillwater Digital Mining Institute, University of the Witwatersrand, Johannesburg 1864, South Africa; mahsan.mahboob@wits.ac.za

[3] Faculty of Engineering, Division of Sustainable Resources, Hokkaido University, 13 jyounishi 8, Kita-ku, Sapporo-City 0608628, Hokkaido, Japan; kawamura@eng.hokudai.ac.jp

* Correspondence: ikeda@gipc.akita-u.ac.jp

**Abstract:** Mining activities often deem mine sites as temporary, leading to their eventual reclamation, rehabilitation, or abandonment. This study innovates by proposing the re-purposing of the disused Osarizawa mine in Akita, Japan, leveraging its consistently low tunnel temperatures to establish a data center, thereby offering a sustainable economic avenue to offset reclamation costs. We assessed the feasibility of this transformation by gathering comprehensive environmental data from the site and conducting meticulous ventilation simulations. These simulations explored various scenarios encompassing diverse ventilation configurations, data server room dimensions, thermal outputs, and the inherent cooling capabilities of the proposed humid rooms. By juxtaposing the simulation outcomes with the criteria set forth in the ASHRAE 2011 Thermal Guidelines, we pinpointed the optimal parameters that satisfy the stringent temperature and relative humidity prerequisites essential for a data center's operation. This research underscores the potential of reimagining abandoned mine sites as strategic assets, providing economic benefits while adhering to critical data center infrastructure standards.

**Keywords:** data center transformation; mine site re-purposing; environmental simulation; sustainable reclamation; ventilation optimization; smart mining

## 1. Introduction

### 1.1. Background

Traditionally, mining is viewed as a temporary land use, the value of mine sites being temporary and primarily tied to the extraction phase. Upon resource depletion or the cessation of economic viability, these sites are often abandoned. The neglect of proper rehabilitation strategies for these sites poses significant environmental risks, such as acid mine drainage (AMD) and contamination from tailings and waste piles [1]. Estimates indicate a significant number of such neglected sites, with figures suggesting around 10,000 in Canada and 5500 in Japan [2]. Recent years have witnessed increased engagement from academic and financial institutions in crafting frameworks for responsible mine closure [3]. Current regulations necessitate the integration of closure and reclamation plans within mining operations, targeting the restoration of landscapes and the mitigation of environmental and socio-economic impacts [4]. While essential, rehabilitation and restoration efforts categorize mine sites as transient land uses, generating no continual revenue but incurring restoration costs. This issue is particularly salient for vast, abandoned underground mines, where complete restoration is often impractical. Re-purposing these

underground spaces provides an opportunity for sustained land use, potentially fostering industries capable of supporting the sites' maintenance and rehabilitation costs [5].

Underground sites offer distinct advantages: they exhibit greater resilience to earthquakes and extreme weather and maintain consistent internal conditions, unaffected by external climate variations [6]. These sites often feature existing infrastructure—like power lines, transport pathways, and ventilation systems—which can be leveraged in re-purposing efforts. However, re-purposing requires continuous monitoring of safety parameters, including communication systems, air quality, and geological stability. Accessibility to these sites is often limited due to their remote locations and depth-related safety challenges. The sites' distances from communities can further complicate re-purposing efforts [6]. Given the access and maintenance challenges of underground spaces, autonomous systems that minimize human intervention are preferred. Implementing a digital twin of the target site enables the use of simulation tools for designing, operating, and monitoring the underground space remotely. This integration not only streamlines operations but also facilitates informed decisions through real-time data, creating a synergistic cyber-physical system (CPS) [5].

### 1.2. Aims and Objectives

This paper aims to study the feasibility of re-purposing the inactive underground Osarizawa mine site located in Akita Prefecture, Japan, into a data center by assessing the environmental conditions of the mine site. This proposal is due to the stable low temperatures of the mine site, that remain at 13 °C all year round, which is advantageous as it has the potential to supply a natural cooling system for the data center. To carry out this study, environmental data such as dry-bulb temperature and relative humidity were collected from the mine site and used to carry out simulations on the mine ventilation simulation program VentSim. The simulation parameters were adjusted to test different scenarios such as different ventilation system arrangements, data server room lengths, server room thermal outputs, and server room–humid room arrangements. The natural cooling capacity of the humid room at different lengths was simulated as well. These parameters were evaluated to study their effects on the dry-bulb temperature and relative humidity of the proposed data server rooms. The energy consumption and operation costs of the fans used for ventilation were also obtained from the simulations.

The simulation results are then compared to the 2011 ASHRAE Thermal Guidelines [7] to assess whether the environmental conditions of the mine site are suitable for the installation of a data center. To aid visualization, and also as a starting point in creating a digital twin, a point cloud of a target area in the underground mine site is created using structure-from-motion (SfM) and multi-view stereo (MVS) photogrammetry. The results of a simulation carried out on this target area are overlaid onto the point cloud to view the results in three dimensions.

## 2. Related Work

### 2.1. Smart Mining

The evolution of digital technologies has paved the way for a new era in mining, known as "smart mining". Smart mining refers to the integrated approach to optimizing mine operations and improving productivity, safety, and environmental impact through real-time data analysis, automation, and advanced technologies [8,9]. One crucial aspect of smart mining is the real-time monitoring and operational control enabled by the internet of things (IoT) and other connected technologies. Ikeda et al.'s (2021) exploration of sensor data communication in underground mining environments highlights the significance of robust and efficient data transmission systems. Their research compares and optimizes multiple installation sequences, contributing to the improvement of real-time data relay and decision-making processes underground [10]. In addition to real-time data monitoring, smart mining also embraces innovative solutions for operational challenges. The work by Ikeda et al. (2019) underscores this aspect through the development of an underground

in situ stress monitoring system using multi-sensor cells and Wi-Fi direct technology [11]. This system not only enhances mining safety by providing immediate stress measurements but also improves operational efficiency. The integration of artificial intelligence (AI) and machine learning (ML) in smart mining operations has further transformed traditional practices. Deep learning techniques, for instance, are being utilized for estimating muckpile fragmentation. Ikeda et al. (2023) employed simulated 3D point cloud data for this purpose, demonstrating the potential of this technology in enhancing ore recovery and reducing waste [9].

Smart mining, therefore, represents a multifaceted approach to modernizing mine operations. By leveraging digital technologies, mining companies can significantly improve operational efficiency, ensure the safety of mine workers, and minimize environmental impact. As this field continues to evolve, further advancements in IoT, AI, and automation are expected to drive the future of mining toward unprecedented levels of optimization and sustainability.

### 2.2. Digital Twin

A digital twin represents a real-time digital counterpart of a physical entity, process, or service. Unlike a prototype, a digital twin autonomously synchronizes data between the physical and digital models, reflecting any changes occurring in the physical model in the digital one. This synchronization is possible through the integration of various digital technologies, including but not limited to, the internet of things (IoT), cloud computing, analytics, and artificial intelligence (AI)—components of Industry 4.0. A digital twin comprises a physical object, its digital model, sensor-acquired time-series data, and the connections between them [8,9,12]. Figure 1 illustrates the structure of a digital twin within a cyber-physical system.

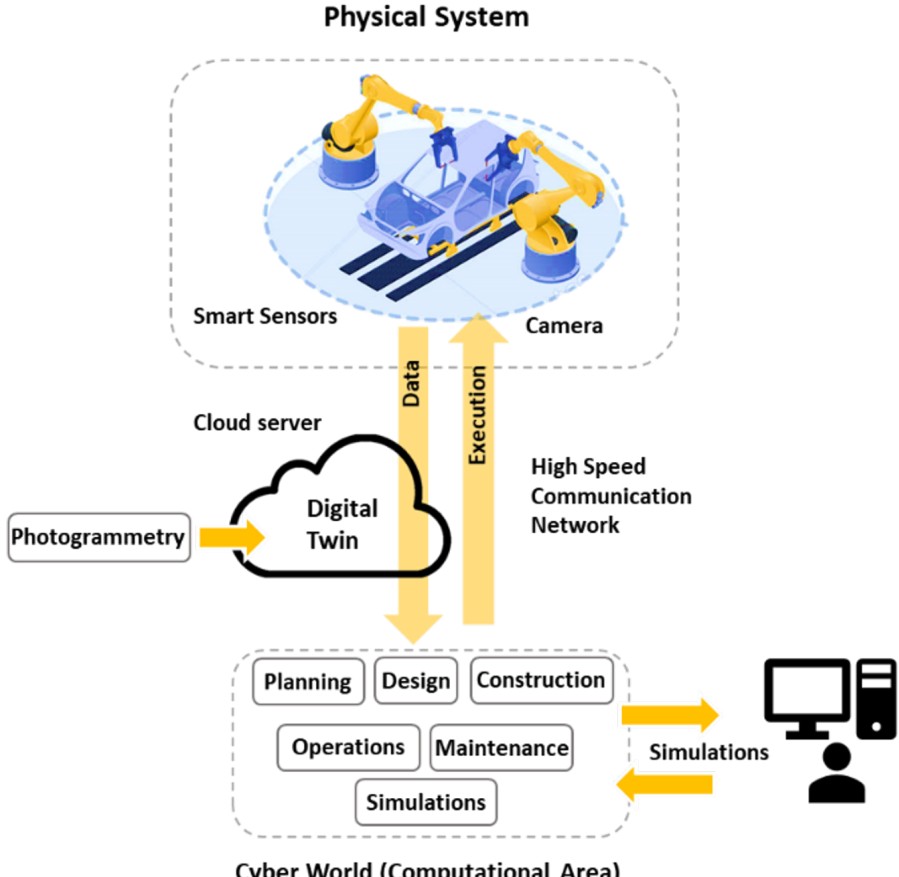

**Figure 1.** Structure of a digital twin.

Data-driven decision making is enhanced by integrating simulation programs with the digital twin, linking physical systems to their digital counterparts. This integration creates a closed-loop known as a digital thread, autonomously updating the virtual model with data from sensors in the physical object. The data informs simulations essential for planning, design, and operations, enabling real-time, informed decision making and risk assessments. Moreover, integrating simulations facilitates the prediction of various outcomes, enhancing operational strategies.

The benefits of utilizing a digital twin include:

- Real-time monitoring and control: With IoT, users can access the digital model of a physical entity or process from virtually anywhere.
- Enhanced safety and efficiency: Automated monitoring of remote or inaccessible areas.
- Predictive maintenance and scheduling: AI-driven solutions detect potential system faults, enabling pre-emptive maintenance and avoiding economic losses or safety hazards.
- Scenario and risk assessment: Simulations with varied input parameters allow the prediction and pre-emptive addressing of unforeseen scenarios.
- Improved documentation: Immediate access to real-time information enhances transparency for stakeholders.
- Streamlined operations: Data-driven decision making, bolstered by advanced analytics and interconnected intelligence networks, fosters synergies and collaborative efforts, boosting productivity.

### 2.3. Ventilation Simulation Programs

Ventilation control is crucial in underground environments, playing a significant role in the operational profitability of underground mining. The planning and optimization of underground ventilation systems demand rigorous analysis, as minor modifications can profoundly impact the entire system. This necessitates the exploration of various options and scenarios, entailing countless predictions related to the ventilation network, system design, and performance. Such complex predictions are unfeasible without the aid of precise and adaptable simulation tools capable of swiftly assessing multiple scenarios [13]. Ventilation simulation programs facilitate the calculation and assessment of environmental variables—such as airflow, temperature, fog, and relative humidity—based on specified parameters like initial temperature, thermal output, and existing ventilation equipment [14].

Previous studies, including those by Webber et al. [14] and Sasmito et al. [15], underscore the necessity of integrating ventilation simulation software for underground mine site monitoring, highlighting the critical role of simulations in enhancing underground systems. For instance, Sasmito et al. [15] demonstrated the efficacy of simulation models in assessing thermal management's impact within mines, focusing on the computational analysis of various thermal factors on underground tunnel airflow. Managing thermal conditions in underground environments is intricate, given the need to consider elements like geothermal gradients and internal heat loads from personnel, lighting, and equipment [15].

This study mirrors the approach of Sasmito et al., albeit with IT equipment supplanting diesel apparatus as the primary heat source [15]. Concurrent research by Kychkin et al. [16] appraises the execution of a TICK stack for IoT applications within a cyber-physical system (CPS) for ventilation, utilizing the InfluxData platform. This research intends to project simulation outcomes onto the digital twin component of the CPS. The VentSim program, chosen for its precise system representation, facilitates the simulation of diverse scenarios for ventilation optimization and transient systems where conditions fluctuate over time [17].

### 2.4. Integrating Simulation Programs with Digital Twins

A well-calibrated digital twin of an underground environment mirrors the actual operational parameters of underground ventilation, incorporating real-time data from sensors within the specified area to offer an updated model of the subterranean conditions. This advanced approach significantly enhances the optimization processes of the ventilation systems. Previous studies have delved into the application of simulation programs for

constructing digital twins, albeit in different contexts. For instance, Dahmen et al. [18] explored this concept not for underground ventilation but for space missions, concentrating on the methodology behind digital twin development and its subsequent utilization for experimentation [14].

Mine ventilation simulation software uniquely presents real-time data gathered from sensors deployed in underground mining locations. This real-time feature not only facilitates continuous monitoring but also autonomously updates the digital twin with fresh data from the mine, a capability explored by Sishi et al. [19]. Similarly, Cheskidov et al. [20] discussed the employment of monitoring systems to display real-time data regarding mining engineering structures, with in situ data analysis programs pinpointing inaccurately measured or calculated characteristics, such as slope safety factors or load-bearing capacities, within a nature-and-technology system. These programs then prompt further necessary measurements and computations. Both the study by Sishi [19] and that by Cheskidov [20] examined the use of digital twins' as real-time monitoring platforms capable of immediate problem identification, yet neither delved into their application as simulation tools.

Jacobs [21] offers an extensive review of the research conducted in this domain, identifying gaps, particularly in enhancing the methodology for digital twin development and its application as a simulation instrument for system evaluation (Table 1). This study aims to fill this gap by employing a digital twin, crafted through photogrammetry, to visualize simulation outcomes. These simulations are designed to mimic the environmental variables of the Osarizawa mine site, assessing its viability as a prospective data center location.

**Table 1.** Different studies discussing integration of digital twin and simulation programs [5].

| Source | Development of Digital Twin | Digital Twin Used to Evaluate a System | Digital Twin Used as a Simulation Tool |
|--------|---------------------------|---------------------------------------|---------------------------------------|
| [22–25] | F | F | F |
| [13,15,26–29] | F | F | T |
| [18,30–35] | T | F | T |
| [14,36–39] | T | F | T |
| [19,20] | F | T | F |

### 2.5. Re-Purposing of Underground Spaces

The re-purposing of a mine site incorporates elements of the existing mining infrastructure and reconfigured landscape aspects for alternative activities post-closure. Such activity not only aids in transitioning the local economy but also mitigates the mine's loss by fostering new forms of attachment to the site and region [6]. An advantage of utilizing underground spaces lies in their stability. The conditions inside underground excavations remain unaffected by external weather conditions and are nearly constant, simplifying temperature and humidity control. Comfortable subsurface temperatures eliminate the necessity for additional insulation, thus conserving energy and reducing costs. Furthermore, underground structures sustain less damage compared to surface structures under earthquake loading [5] and possess increased resistance to hurricanes, tornadoes, and most weapon system penetrations.

However, obstacles exist in re-purposing mine sites, including their susceptibility to flooding and the requirement for diverse skill sets. The location also poses a challenge, as mine sites are often situated in remote areas, far from communities and towns, leading to minimal community involvement that could otherwise motivate re-purposing. The stability and utility of these spaces must be continually re-evaluated and maintained throughout the re-purposing process to ensure safety. The geothermal gradient may cause temperatures inside the mined cavity to increase with depth, necessitating the utilization of pre-existing mine cooling systems for deep mine sites. Globally, examples of re-purposing are few compared to the number of closed mines. Out of the 1804 inactive mine sites documented

in the S&P database, only approximately 141 operations have undergone some form of re-purposing [6].

It is common for a mine site to serve multiple purposes after mining; post-mining transitions frequently encompass various categories of land use. Factors such as proximity to communities, infrastructure connectivity, and company policies also influence the re-purposing of mined underground spaces. The most prevalent re-purposing activities include wildlife habitats, historical museums or exhibitions, cultural or historical precincts, and parks or open green spaces [6]. Table 2 illustrates other instances of mine re-purposing, like healthcare facilities such as Solotvyno's Allergological Hospital in Ukraine, which specializes in treating patients with respiratory ailments. Figure 2 depicts underground agriculture employing LED lights and hydroponic technology, while Figures 3 and 4 present examples of a former salt mine in Germany, re-purposed for radioactive waste storage, and a former limestone mine, converted into a recreation center, respectively. With technological advancements and enduring demand for metals and other resources, the depth of underground mining sites is anticipated to increase. Re-purposing deeper mines will require considering the geothermal gradient—whereby underground mine temperatures rise with depth—and leveraging previous mine site cooling systems to maintain optimal temperatures.

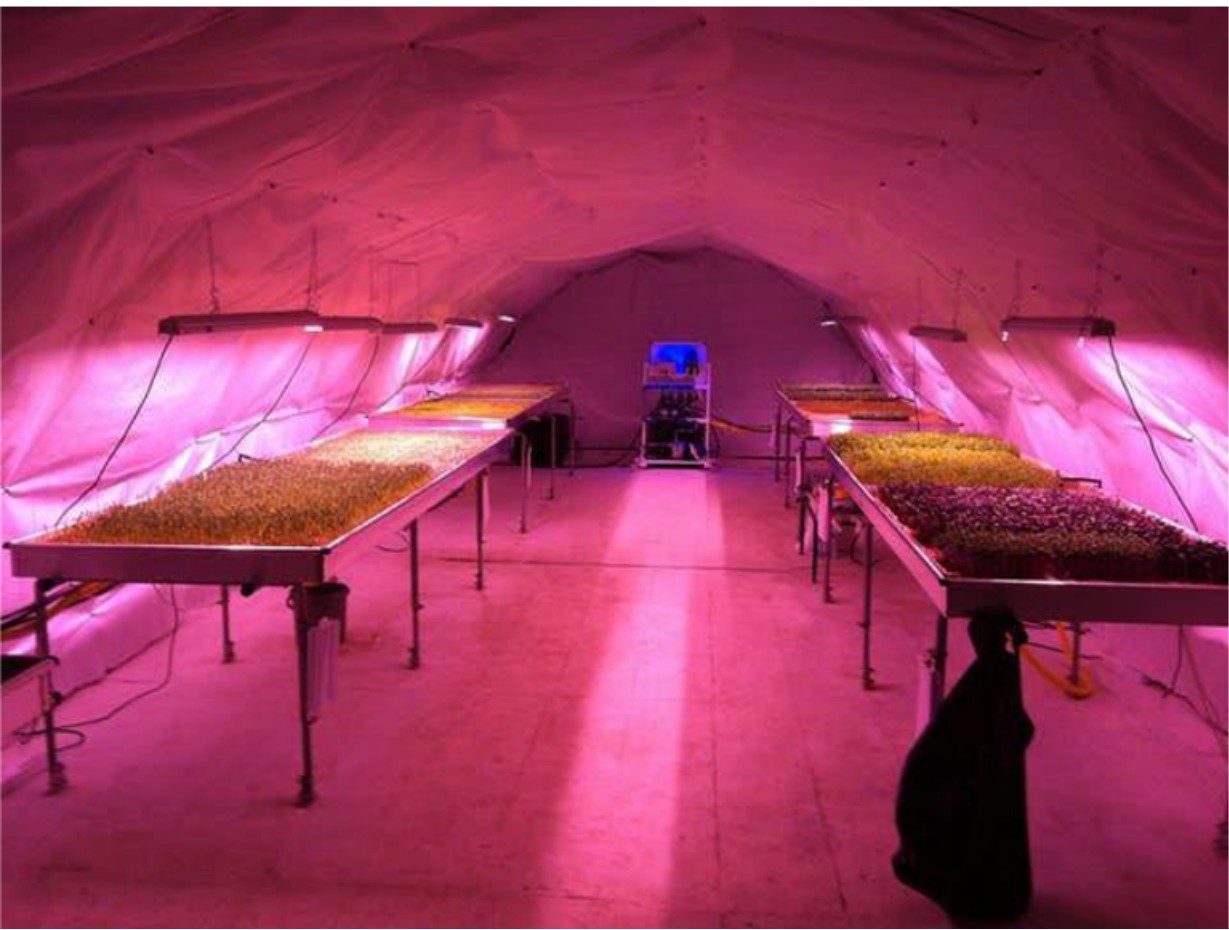

**Figure 2.** Blast design at target site.

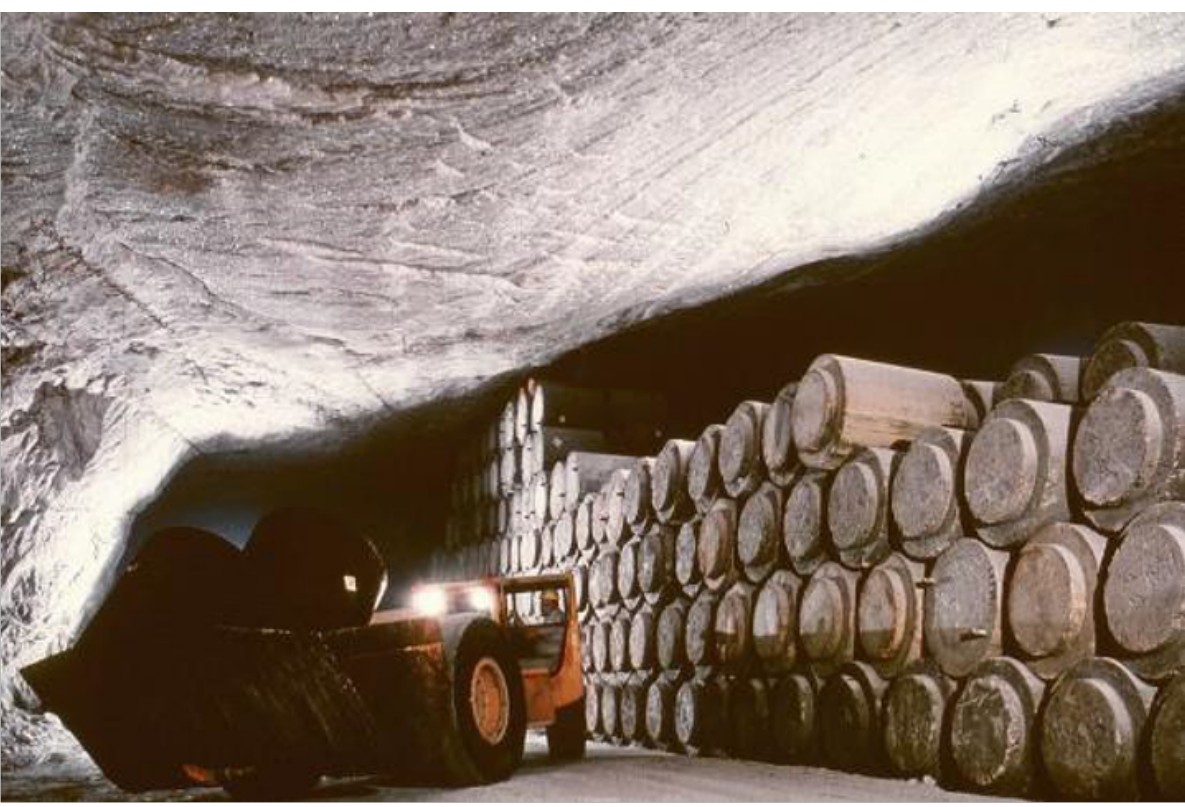

**Figure 3.** Drums with concrete shielding to reduce radiation exposure stored in Asse II, previously Asse salt mine. Adapted from [37].

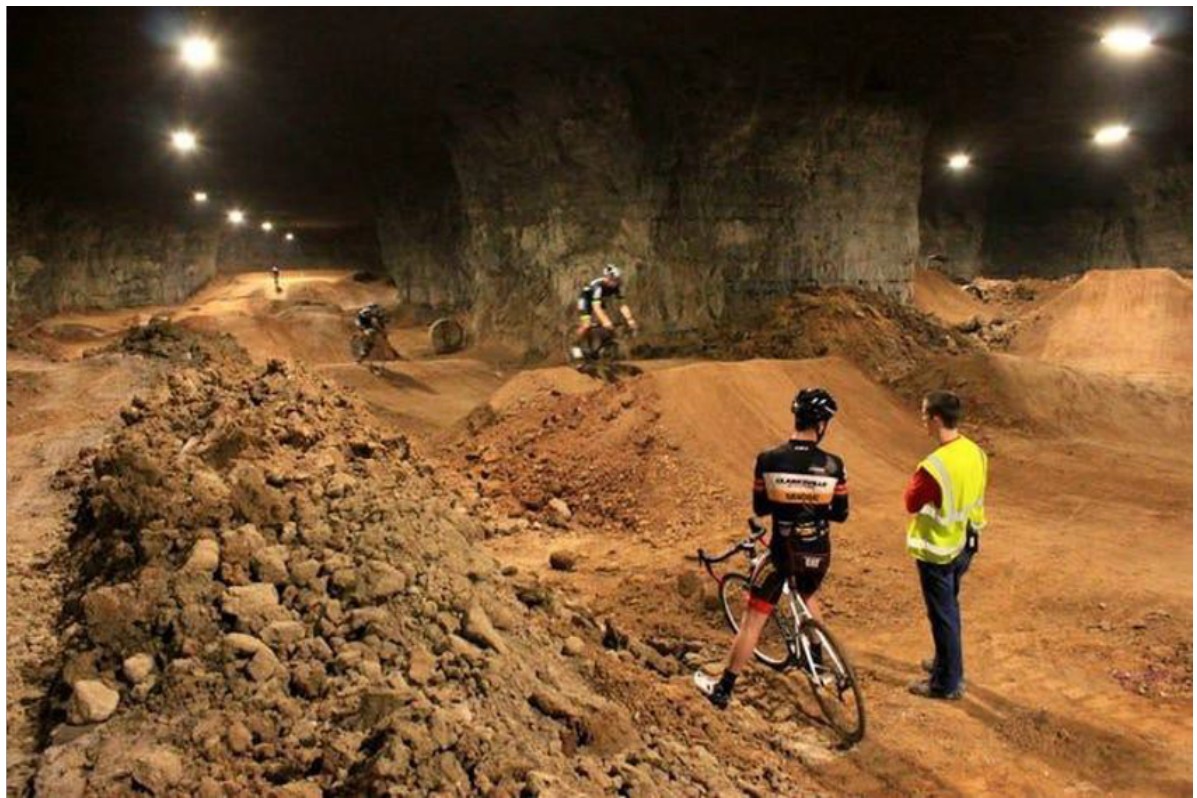

**Figure 4.** Former limestone mine in Louisville, Kentucky, restructured to create underground bike track. Adapted from [38].

**Table 2.** Examples of mine re-purposing.

| Function | Details | Depth | Mine Name |
|---|---|---|---|
| Underground laboratory | Stawell underground physics lab<br>Kamioka underground laboratory<br>(Japan, 1982) | −700 m<br>−1000 m | Stawell Victoria gold mine<br>Kamioka Mozumi arsenic mine |
| Waste disposal<br>Geothermal exploration<br>Industrial tourism<br>Healthcare | Asse II mine (Germany, 1967)<br>Underground hot water circulatory system<br>(Canada, 1980s)<br>Kailuan national mine park (China, 2008)<br>Solotvyno's Allergological Hospital<br>(Ukraine, 1965) | −750 m<br>−2000 m<br>−410 m<br>−300 m | Asse salt mine<br>Hot spring coal slope mine<br>Tangshan coal mine<br>Solotvyno's salt mine |
| Leisure and community | Louisville Mega Cavern—underground bike<br>park (USA) | −30 m | Mega cavern former limestone mine |

*2.6. Data Centers*

Data centers serve as pivotal infrastructure where organizations house critical applications and data. The paradigm has transitioned from traditional on-premises servers to sophisticated virtual networks, facilitating operations in multi-cloud environments [40]. These centers underpin a myriad of applications and activities, encompassing virtual desktops, enterprise resource planning, and advanced realms like artificial intelligence and big data analytics. Figure 5 illustrates the basic components in a data center facility, highlighting their complex architecture.

The year 2020 witnessed a 40% upsurge in global internet traffic, propelled by a proliferation in video conferencing, online gaming, and digital connectivity [40]. Forecasts suggest that by 2024, remote work will be integral to over 90% of infrastructure operations, and centralized platform engineering will be embraced by more than half of enterprises by 2025 [41]. This digitalization trajectory accentuates the imperative for robust data accessibility and security protocols, challenging to internalize within organizations.

Power reliability is critical, with outages having substantial financial repercussions, potentially costing up to USD 17,244 per minute [42]. Re-purposing underground mine sites as data centers presents a viable alternative, capitalizing on existing infrastructural elements like power lines and backup generators, essential for uninterrupted power supply (UPS) systems. The proposition for re-purposing the Osarizawa mine site underscores this potential.

Physical security complements cyber security in its importance, prompting some data centers to utilize underground facilities' inherent security advantages [40]. Examples include the Lefdal Mine Datacenter in Norway and the Bluebird Underground Data Center in Missouri, re-purposed from mining operations, offering natural cooling and fortified protection [43,44]. Figure 6 shows the Lefdal Mine Datacenter, demonstrating its robust security features including a single point of entry.

Innovative approaches extend to Microsoft's underwater data center, Project Natick, and proposed extraterrestrial data centers, aiming to exploit the natural cooling properties of these environments to reduce energy consumption [45,46]. Figure 7 captures the retrieval of Microsoft's Northern Isles data center (Project Natick) in July 2020, showcasing its unique underwater environment.

The integration of renewable energy sources and the conceptualization of data centers as heat sources for neighboring infrastructure further enhance their sustainability [47]. This research advocates for leveraging the Osarizawa mine's ambient temperatures, promoting energy efficiency, and operational cost reductions in prospective data center applications.

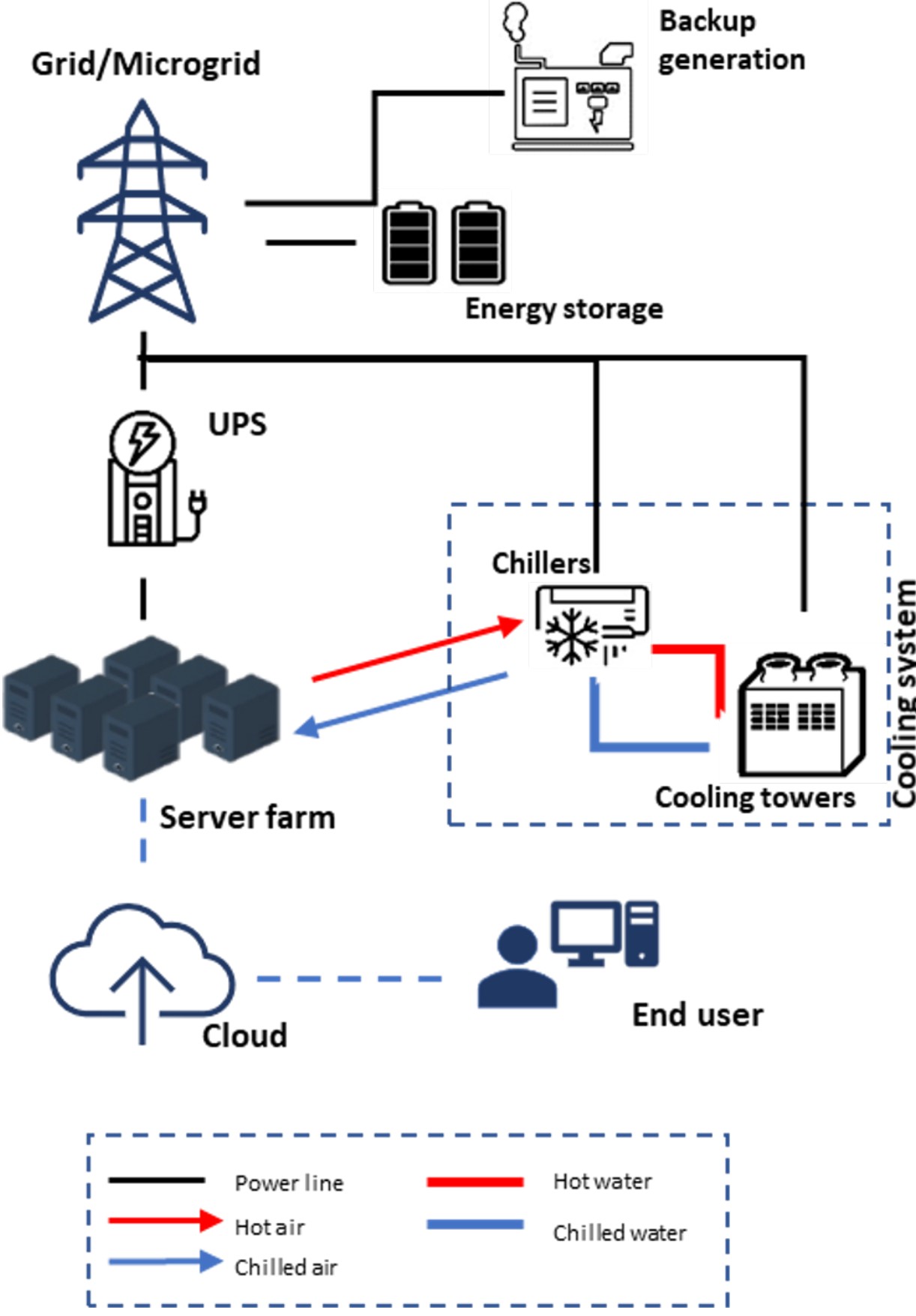

**Figure 5.** Basic components in a data center facility.

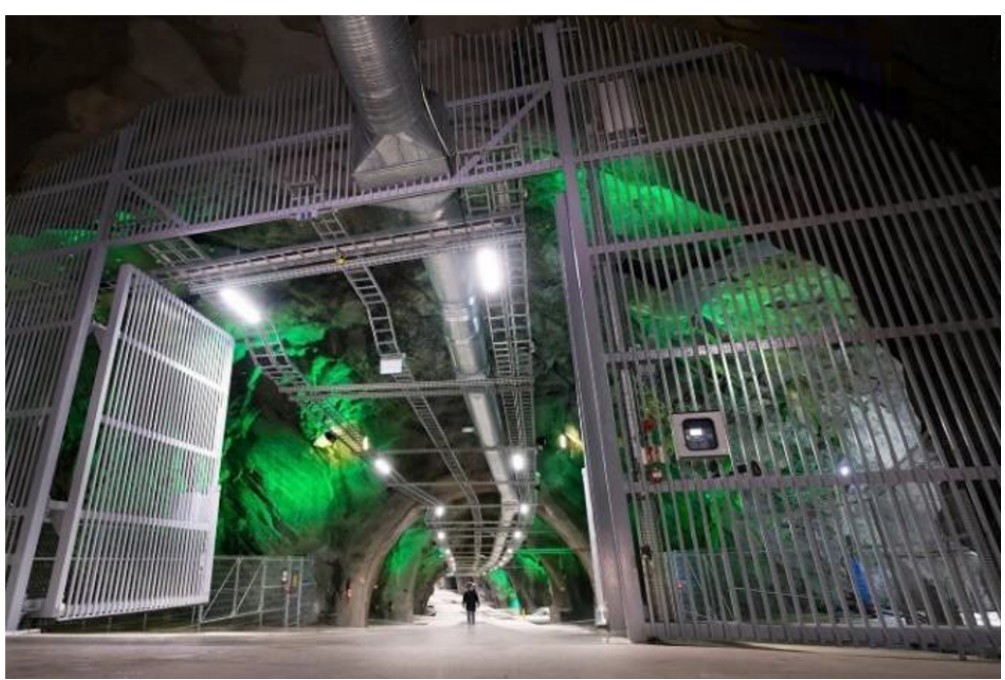

**Figure 6.** Lefdal Mine Datacenter has a single point of entry [24].

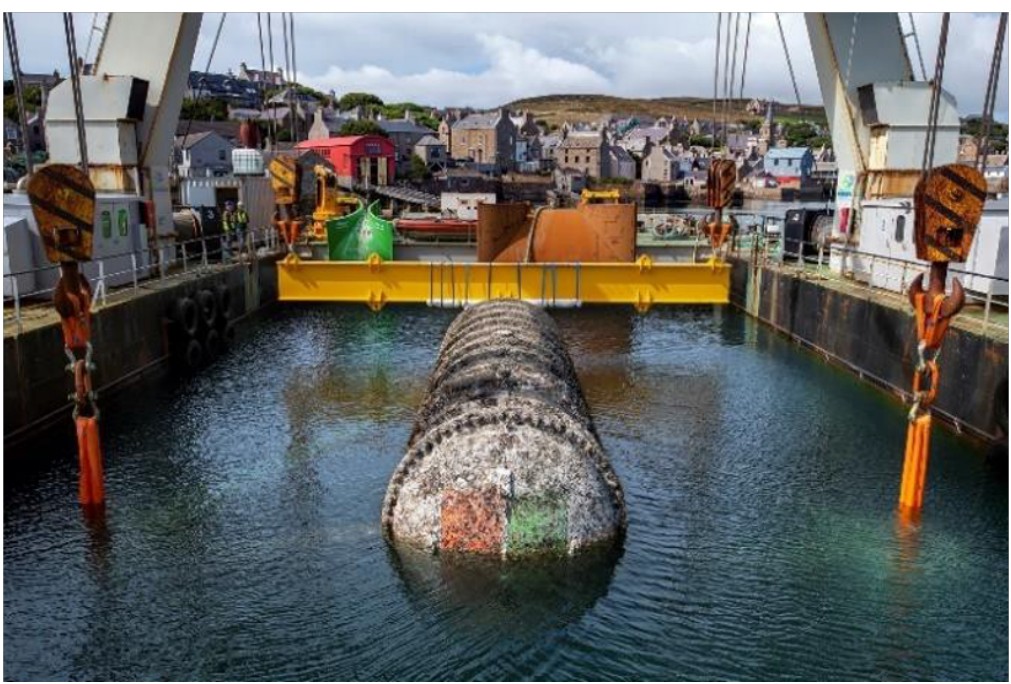

**Figure 7.** Retrieval of Microsoft's Northern Isles data center (Project Natick) in July 2020.

*2.7. ASHRAE Thermal Guidelines for Data Centers*

The American Society of Heating, Refrigerating, and Air-Conditioning Engineers (ASHRAE) established a standard for data processing environments with the release of the first edition of the *"Thermal Guidelines for Data Processing Environments"* in 2004, introducing generic server equipment metrics [7]. This seminal document aimed to harmonize the environmental conditions conducive to optimal data center operations, which had previously been subject to varying specifications by different information technology equipment (ITE) manufacturers. The parameters adopted in the current study are derived from the 2011 ASHRAE thermal guidelines [7].

ASHRAE's guidelines primarily focus on two critical environmental variables: dry-bulb temperature (DBT) and relative humidity (RH). DBT refers to the air temperature measured by a thermometer freely exposed to the air, while RH represents the ratio of the actual vapor pressure relative to the saturation pressure at the DBT. ASHRAE delineates environmental envelopes, characterized by specific RH and DBT ranges, suitable for IT equipment. These envelopes are categorized into various classes, with classes A1, A2, and A3 tailored for data centers, class B designated for office or residential use, and class C for industrial applications, as outlined in Table 3. Figure 8 depicts the recommended environmental envelope in a psychrometric chart, with the recommended and allowable DBT and RH for each class detailed in Table 4 [7].

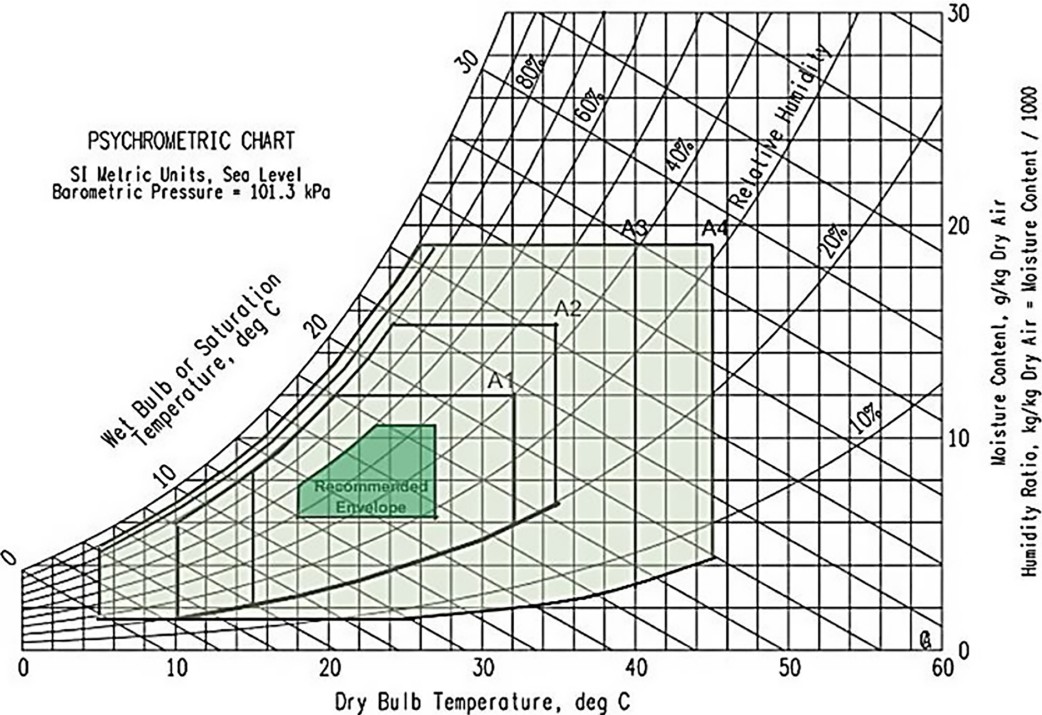

**Figure 8.** Psychrometric chart with recommended envelope (center, green) and allowable envelope (outer, light green) [7].

**Table 3.** ASHRAE IT equipment classification [7].

| Class | Application | IT Equipment | Environmental Control |
|-------|-------------|--------------|-----------------------|
| A1 | | | Tightly controlled |
| A2 | Data center | Enterprise servers, storage products, personal computers, workstations | Some control |
| A3 | | | Some control |
| A4 | | | Some control |
| B | Office/Home | Personal computers, workstations, laptops, printers | Minimal |
| C | Industrial | Ruggedized controllers, PDAs | No control |

**Table 4.** 2011 ASHRAE Thermal Guidelines for Data centers

| Class | Dry-Bulb Temp. (°) | Humidity Range (%) | Max Dew Point (°) |
| --- | --- | --- | --- |
| A1 | 15–32 (5–45) * | 20–80 (8–80) * | 17 (27) * |
| A2 | 10–35 (5–45) * | 20–80 (8–80) * | 21 (27) * |
| A3 | 5–40 (5–45) * | 8–85 (8–85) * | 24 (27) * |
| A4 | 5–45 (5–45) * | 8–90 (8–90) * | 24 (27) * |
| B | 5–35 (5–45) * | 8–80 (8–90) * | 28 (29) * |
| C | 5–40 (5–45) * | 8–80 (8–80) * | 28 (29) * |

* Values in parentheses represent the product power off condition.

The pivotal role of efficient ventilation and cooling systems in data centers stems from the necessity to manage the substantial thermal output produced by high-density IT equipment operating continuously. Ineffective thermal management can escalate energy consumption and may expose IT equipment to conditions surpassing the permissible environmental envelopes, potentially resulting in hardware damage or data loss [48]. Such disruptions not only necessitate expensive repairs or replacements but also culminate in significant productivity and revenue deficits.

Prolonged exposure of IT equipment to elevated levels of RH can precipitate corrosive damage and media tape errors. Conversely, environments with low RH foster conditions conducive to electrostatic discharge (ESD) events, characterized by the abrupt equalization of differing electrical potentials between two objects. ESD can compromise semiconductor devices, irreparably damage IT equipment, and, in severe instances, trigger fires or explosions.

### 3. Methodology

#### 3.1. Collecting Data from Osarizawa Mine Site

Data in the form of images and temperature, relative humidity readings were collected from 10 m outside the entrance of the Osarizawa tourist mine site to approximately 500 to 600 m from the entrance. Images were captured using an Insta360 camera and subsequently processed through the Structure-from-Motion Virtual Multi-View Stereo (SfM-VMS) method to generate a point cloud. These data were utilized to construct a mesh, thereby creating a digital twin of the targeted tunnel section, specifically 650 to 750 m from the entrance. The mining ventilation simulation software, VentSim (Version 5.4.4.8), was employed to emulate the natural cooling phenomena of a humid room situated within a server room-humid room-server room configuration, leveraging the data procured from the designated area of the Osarizawa mine site. The thermal output of the data center was computed and escalated in proportion to the elongation of the server room tunnel.

Simulations were conducted to investigate the impacts of varying server room lengths, disparate airflow volumes, and assorted humid room lengths on the temperature and relative humidity within the data server rooms. The outcomes from these simulations were subsequently evaluated in accordance with the ASHRAE 2011 Thermal Guidelines [7] to ascertain their compliance with data center prerequisites. Finally, the simulation results were superimposed onto the fabricated digital twin utilizing the CloudCompare software, enhancing the visualization of temperature and relative humidity distribution across the data servers and humid rooms. Figure 9 shows the inside of the tourist mine tunnel. Figure 10 shows a map of the tourist mine site layout.

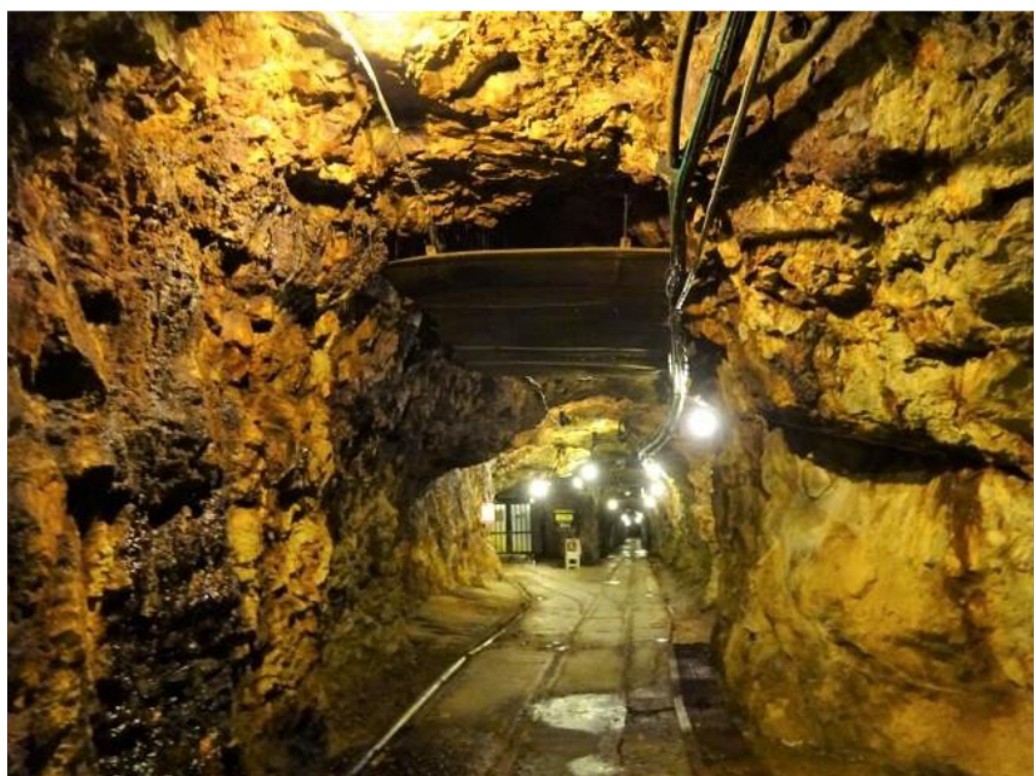

**Figure 9.** The inside of the Osarizawa tourist mine tunnel.

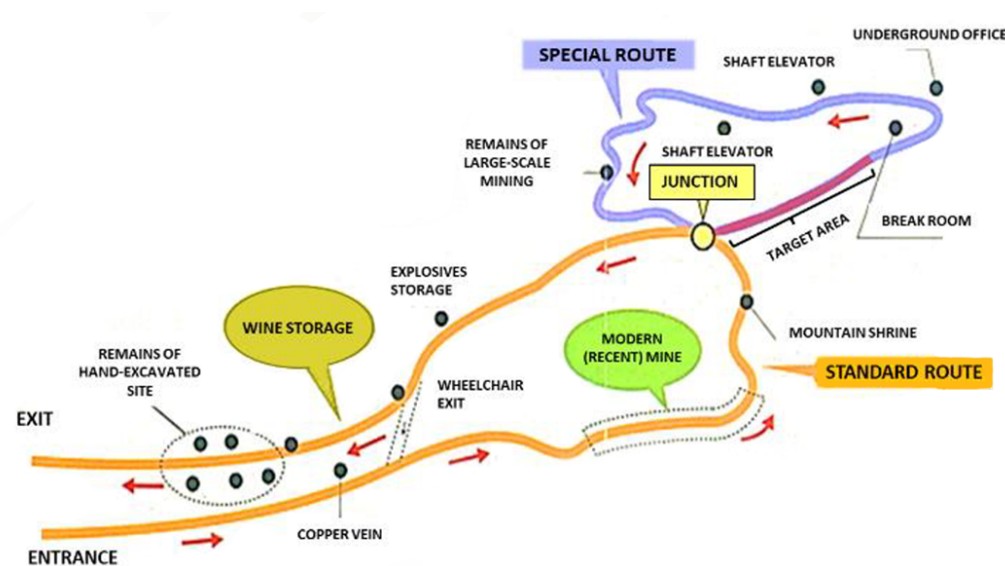

**Figure 10.** The layout of the Osarizawa tourist mine site. Entrance to tunnels beyond this site were prohibited.

### 3.1.1. Interview with Osarizawa Mine Site Staff

An interview was conducted via telephone with a staff member from the Osarizawa mine site's tourist center. The discussion focused on various aspects such as the humidity, the fraction of wet rock, the position of the groundwater table relative to the excavated tunnels, and the temperature within the tunnels extending beyond the tourist mine. It is noteworthy that the underground tunnels at the Osarizawa mine site stretch up to 800 km from the entrance. However, the tunnels beyond the tourist area are administered by a different entity, the Eco-Management Corporation (Osarizawa branch), a subsidiary of

Mitsubishi Materials Corporation. The information sought is considered confidential and is not available for public disclosure. Consequently, data were gathered exclusively from the tunnels accessible within the tourist mine site, which cover a distance of 2.1 km. The simulation input parameters, including humidity, dry-bulb temperature, tunnel dimensions, and rock wetness fraction, were predicated on the assumption that the conditions in the sections of the mine beyond the tourist area do not significantly deviate from those observed within the tourist-accessible areas.

### 3.1.2. Collecting Data from Tunnel Section

The relative humidity and dry-bulb temperature data were collected at the Osarizawa tourist mine site, starting 10 m outside the tunnel entrance and subsequently every 10 m up to 1.1 km from the entrance. The target tunnel section was defined from 650 m to 750 m, representing the farthest accessible area from the entrance. The tunnel walls were observed to be moist, with certain areas saturated enough to support moss growth. The temperature recorded 10 m outside the tunnel was 19°, decreasing to 12° before stabilizing approximately 200 m into the tunnel. To minimize interference from external weather conditions, measurements were primarily utilized from points around 500 m from the entrance. Figure 11 displays a thermal image captured with a FLIR camera, illustrating the temperature variation at the tunnel entrance. High-resolution images of the target tunnel were obtained using the Insta360, a 360-degree camera, as shown in Figure 12. Employing a 360-degree camera enhances the resolution by minimizing human error, as even slight discrepancies in the position, height, and orientation of the captured images can affect the subsequent image processing quality.

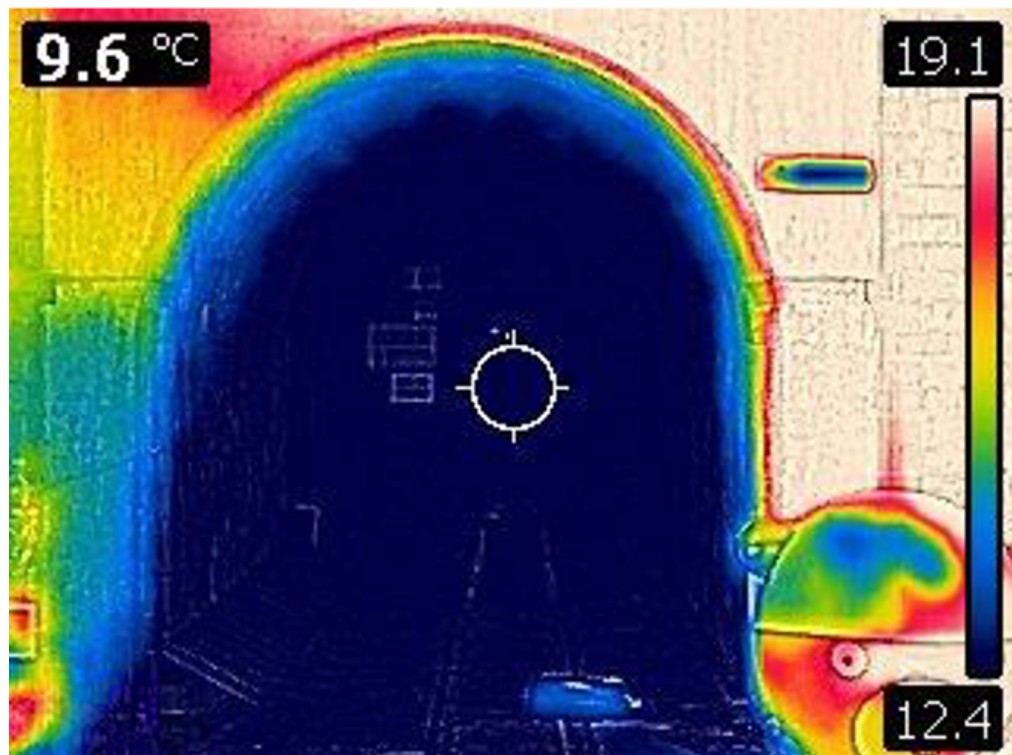

**Figure 11.** Temperature decreases from 19° to 12° at tunnel entrance.

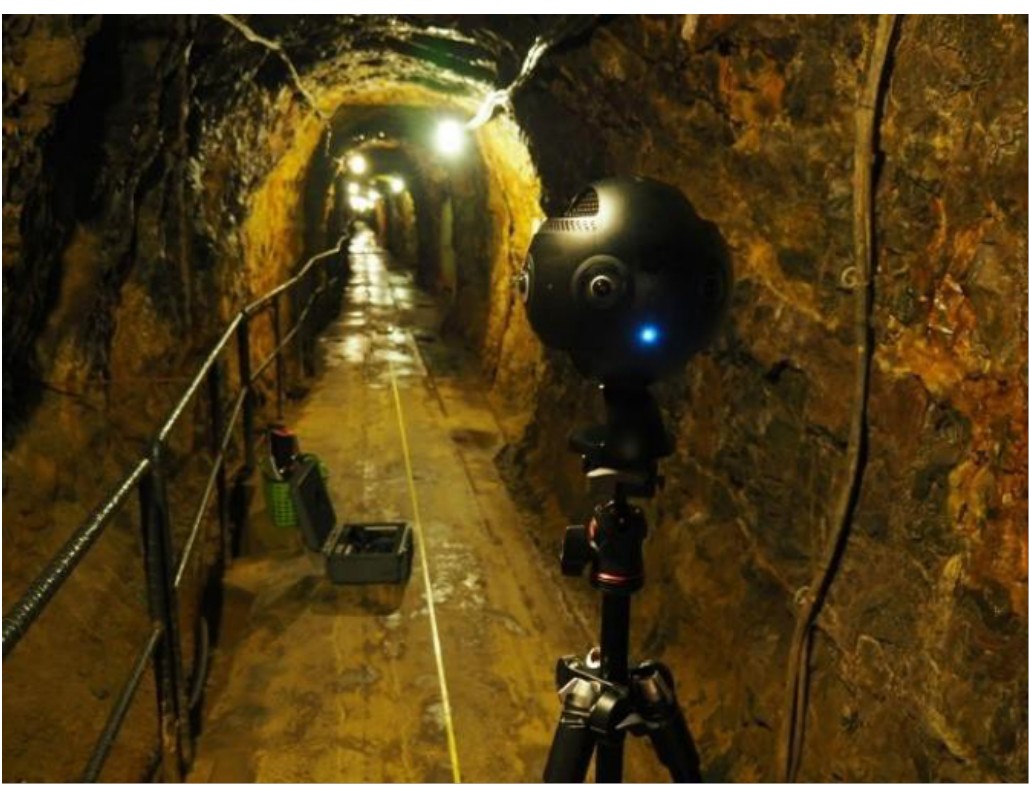

**Figure 12.** Using Insta360 to collect image samples.

The resolution and texture of the images significantly influence the number of key points extracted, with high-quality, original resolution images yielding optimal results. The conditions from the target area to the exit were presumed to be consistent with those from the target area to the entrance. At each measurement point, the temperature and relative humidity at heights of 0 m, 1 m, 2 m, and 3 m (for areas where the tunnel height exceeded 2.5 m) above the ground were recorded to verify the absence of a pronounced temperature gradient within the tunnel. Table 5 illustrates the dry-bulb temperature and relative humidity of the target tunnel section from 650 m to 750 m, measured at various heights (shown in Table 5). The discrepancies in the relative humidity and dry-bulb temperature across different heights were less than 1°, hence deemed to be negligible.

**Table 5.** Target section (650 m to 750 m) of tunnel: dry-bulb temperature and relative humidity.

| Distance (m) | Dry-Bulb Temperature (°) | | | Relative Humidity (%) | | |
|---|---|---|---|---|---|---|
| Height (m) | 0 | 1 | 2 | 0 | 1 | 2 |
| 650 | 13.1 | 13.1 | 13.2 | 95.4 | 95.4 | 95.4 |
| 660 | 13.0 | 13.0 | 13.0 | 95.4 | 95.3 | 95.4 |
| 670 | 13.0 | 13.0 | 13.0 | 95.2 | 95.3 | 95.3 |
| 680 | 13.0 | 13.0 | 13.0 | 95.3 | 95.3 | 95.3 |
| 690 | 12.9 | 12.9 | 12.9 | 95.4 | 95.4 | 95.8 |
| 700 | 13.2 | 13.2 | 13.2 | 94.2 | 94.5 | 94.2 |
| 710 | 13.1 | 13.1 | 13.1 | 94.7 | 94.1 | 94.1 |
| 720 | 13.1 | 13.1 | 13.1 | 94.0 | 94.5 | 94.5 |
| 730 | 13.0 | 13.0 | 13.1 | 94.9 | 94.9 | 95.0 |
| 740 | 13.1 | 13.1 | 13.2 | 95.1 | 95.1 | 95.2 |
| 750 | 13.5 | 13.4 | 13.4 | 95.1 | 95.4 | 95.0 |

### 3.2. Image Processing Using SfM-MVS Process

Photogrammetry was utilized to create a digital twin of the target tunnel section. The digital twin facilitated the visualization of the simulated results by color-adjusting

the points in the point cloud to represent temperature and relative humidity. The images captured using the Insta360 were processed through structure-from-motion (SfM) and multi-view stereo (MVS) methods to generate a point cloud and mesh, resulting in the digital twin of the tunnel section spanning 650 m to 750 m.

SfM is a photogrammetric range that employs algorithms to identify corresponding points in overlapping images, crucial for 3D models derived from 2D images captured from various angles [49]. The initial phase of image processing in SfM is feature detection, which involves identifying potential joint features in each image. The scale-invariant feature transform (SIFT), known for its resilience to image pixel noise, is commonly employed for this purpose [50].

SIFT extracts points with consistent scaling, and rotation, and those invariant to lighting changes and 3D camera view angles. Each key point is assigned a unique descriptor. These key points undergo a matching process to find correspondences using the descriptors. This step includes fitting key points to their nearest neighbor based on the least Euclidean distance for its descriptor vector, while unmatched keypoints are discarded (Figure 13).

The subsequent stage, sparse reconstruction or structure-from-motion, converts 2D images into a sparse 3D structure. This method also reconstructs the 3D scene structure, camera pose, orientations, and calibration parameters. The process comprises initialization, image registration, triangulation, and bundle adjustment, culminating in a precise reconstruction, as depicted in Figure 14.

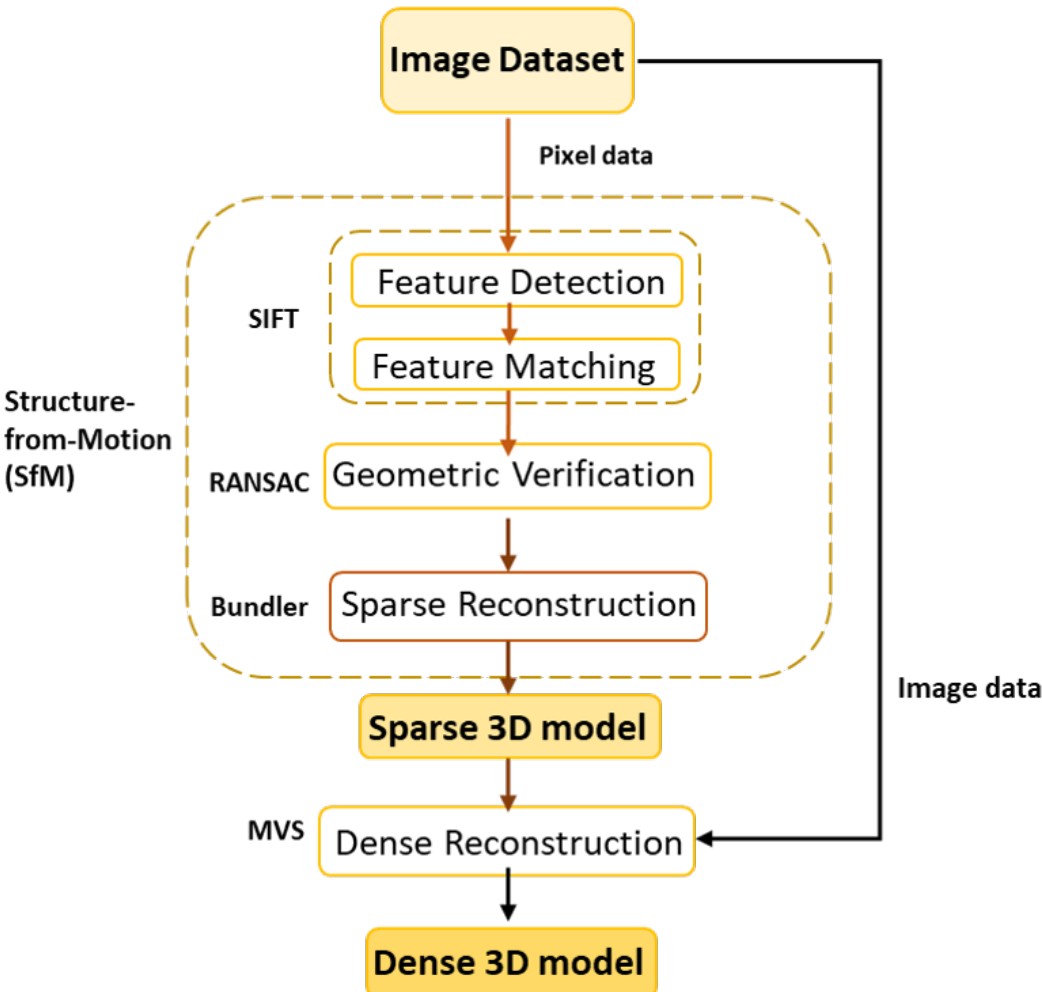

**Figure 13.** General workflow of SfM-MVS process.

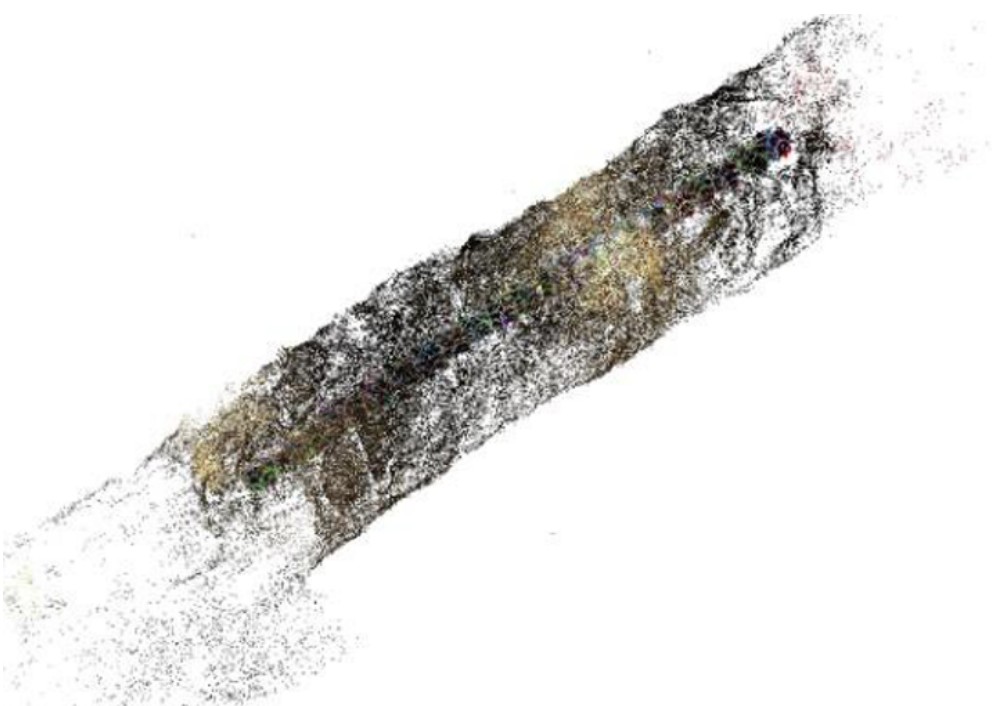

**Figure 14.** Results of sparse reconstruction.

The final phase involves applying the multi-view stereo process, where the clustering views for MVS (CMVS) and patch-based MVS (PMVS) are utilized for generating a "dense" 3D model [51]. The outcome of this dense reconstruction is illustrated in Figure 15.

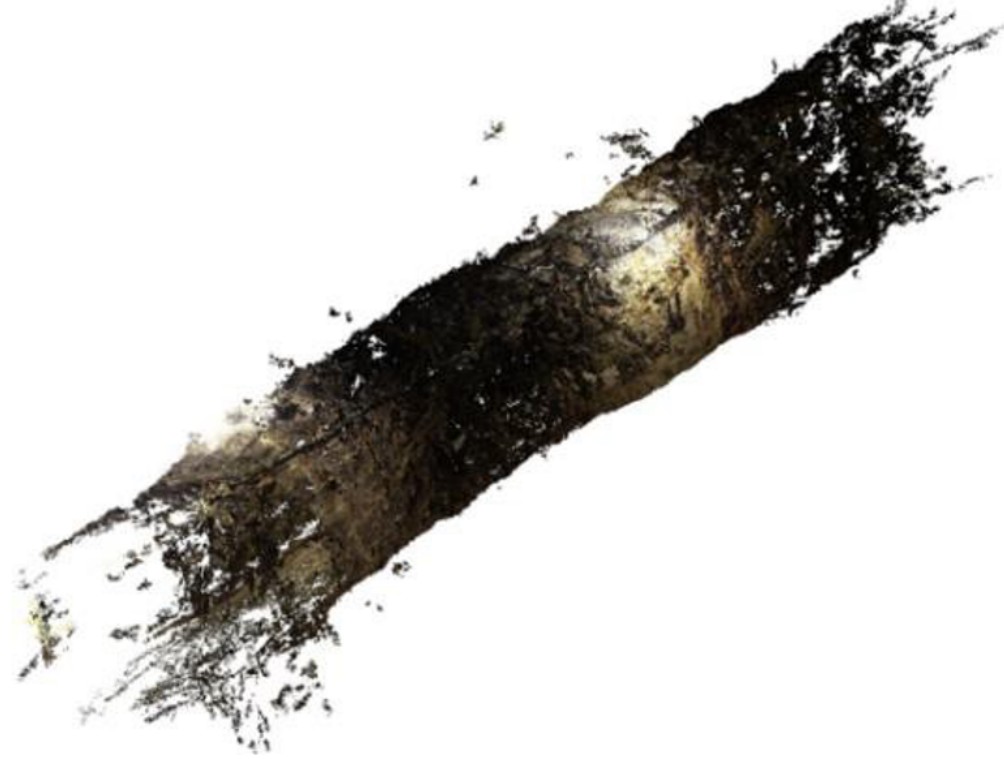

**Figure 15.** Results after dense reconstruction CMVS-PMVS.

*3.3. Calculating the Thermal Output of Data Center to Be Simulated in Programs*

The computation of a data center's cooling system necessitates an estimation of the center's total thermal output. The heat generated from computing or other IT equipment

processes is often insubstantial; hence, the electricity drawn from the AC power mains essentially converts entirely into heat. Consequently, the thermal output of a data center equates directly to its power input [52]. Calculating the total thermal output requires considering the heat contributions from various sources, including IT equipment, UPS systems, power distribution units, air conditioning units, lighting, and personnel. Rasmussen [52] offers a practical worksheet for estimating a data center's thermal output, as shown in Table 6. This approach employs simple rules yielding results within the acceptable margin of error compared to more complex analyses that account for every item's thermal contributions within the center. Notably, the thermal outputs of UPS and power distribution systems for IT equipment are consistent across different brands, permitting accurate approximations. Similarly, the thermal outputs from lighting and personnel can be readily estimated with primary considerations being the data center's floor area and rated electrical system power.

Considering a data center occupying 100 m$^2$ with a rated power of 240 kW, comprising 150 racks and a maximum staff of 10, the thermal output calculation follows the guidelines provided by Rasmussen et al. [53], which suggest a typical data center operates at about 30% capacity. Thus, the IT equipment's total thermal output amounts to 30% of 240 kW. Given the data center's subterranean location, heat gain from solar radiation through windows and doors is negligible.

**Table 6.** Quick estimation worksheet for data center thermal output

| Item | Data Required | Heat Output Calculation |
| --- | --- | --- |
| Equipment | Total load power (W) | Equal to total load power |
| UPS with Battery | Power system rated power (W) | $(0.04 \times$ Power system rating$) + (0.05 \times$ Total load power$)$ |
| Power Distribution | Power system rated power (W) | $(0.01 \times$ Power system rating$) + (0.02 \times$ Total load power$)$ |
| Lighting | Floor area (either in ft$^2$ or m$^2$) | $2.0 \times$ floor area (ft$^2$) or $21.53 \times$ floor area (m$^2$) |
| People | Maximum number of personnel | $100 \times$ max number of personnel |
| **Total** | Subtotals from above | Sum of heat output subtotals |

Based on the worksheet in Table 6 and assuming (a) a total IT load of 240 kW, and (b) a maximum staff of 10, the total thermal output for the data center is estimated at approximately 92,193 kW, as detailed in Table 7. This figure is simplified to 92 kW for practical purposes and serves as the input parameter for a data center tunnel of 100 m length, scaling proportionally with increases in the server room tunnel length (e.g., a 200 m server room equates to 184 kW, etc.).

**Table 7.** Estimate of data center/server room thermal output at 240 kW total IT power load.

| Item | Heat Output Calculation (W) | Heat Output Subtotal (W) |
| --- | --- | --- |
| Equipment | 30% of 240,000 | 72,000 |
| UPS with Battery | $(0.04 \times 240{,}000) + (0.05 \times 72{,}000)$ | 13,200 |
| Power Distribution | $(0.01 \times 240{,}000) + (0.02 \times 72{,}000)$ | 3840 |
| Lighting | $21.53 \times 100$ | 2153 |
| People | $100 \times 10$ | 1000 |
| **Total** | | 92,193 |

### 3.4. VentSim Analysis

The mining ventilation simulation program VentSim was utilized to emulate the natural cooling effects in the server rooms by varying the length of the humid room tunnels. Diverse configurations of server and humid rooms, varying fan airflows, as well as maximum server room lengths, were subjected to simulation. The resultant dry-bulb temperatures and relative humidity levels within the server room were then juxtaposed with the ASHRAE 2011 Thermal Guidelines [7] to determine compliance with the recommended environmental envelope. This paper simulated two distinct arrangements and

scenarios. In both cases, arched tunnels with parameters of 4 m width and 5 m height were employed, with the tunnel lengths modified in increments of 100 m. Damp-proof walls were instituted to segregate the server rooms from the humid rooms. Refer to Figure 11 for a visual representation of the tunnel configurations.

### 3.4.1. Free Cooling Provided by Humid Sections

Free cooling, also known as the economizer cycle, leverages the natural climate to cool data centers, providing an alternative to traditional cooling mechanisms such as air conditioning. There are three primary methods of free cooling: airside, waterside, and heat pipe free cooling. An exemplary initiative is Microsoft's Project Natick, which utilizes the surrounding seawater for comprehensive site-wide waterside free cooling. Zhang et al. [46] compiled an extensive list of data centers that employ innovative and emblematic free cooling systems, highlighting their energy conservation and cooling capabilities. This research explores the feasibility of airside free cooling for a data center, capitalizing on the consistently low ambient temperature of 13° within the Osarizawa mine site.

Our study employs this design, utilizing an exhaust fan to propel cool air into the server room, while the heated air is redirected into an adjoining humid room, following the airflow sequence. The primary objective of this research is to simulate the potential of a humid segment, situated between successive server rooms, functioning as a natural cooling mechanism for the subsequent server room. The proposed layouts for this system are depicted in Figures 16 and 17. In the absence of persistent airflow, leading to air stagnation, the conditions of both rooms would seek equilibrium, resulting in high-humidity air migrating into the lower-humidity server room. This sudden influx can induce condensation, posing a risk to the IT equipment. Therefore, it is crucial, in instances of power disruptions or fan malfunctions, that this aperture is sealed to prevent the incursion of moisture from the humid room into the server room.

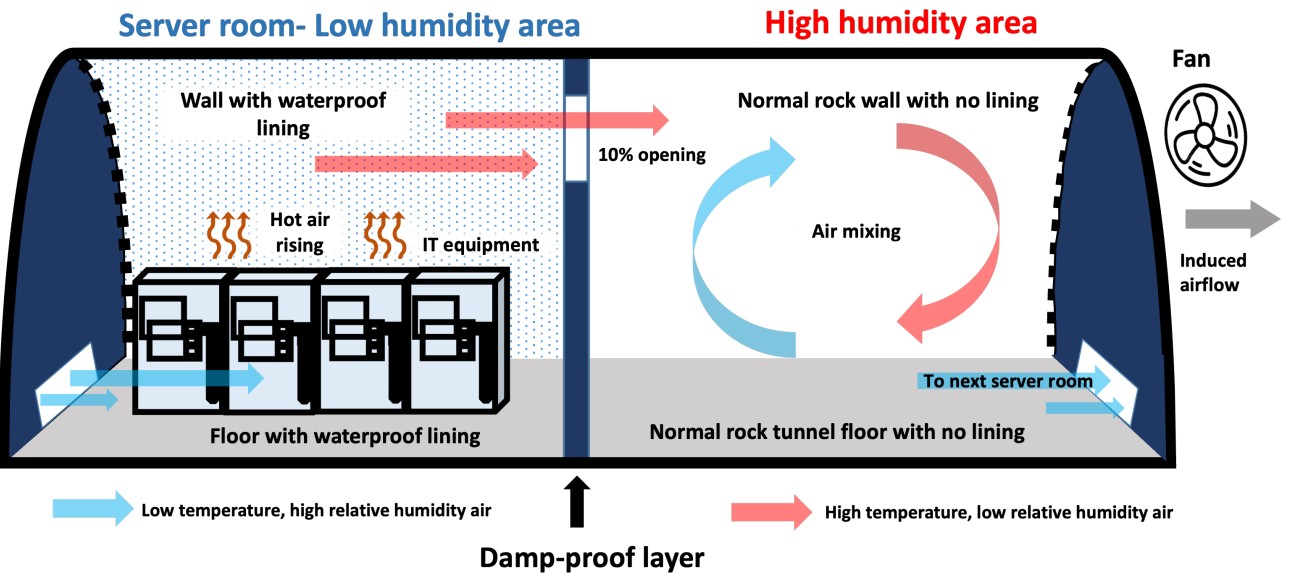

**Figure 16.** The layout of tunnels simulated in scenario 1. Left is the server room (low-humidity area) while the section on the right is the humid room (high-humidity area). Openings and an exhaust fan are simulated to induce airflow between the sections.

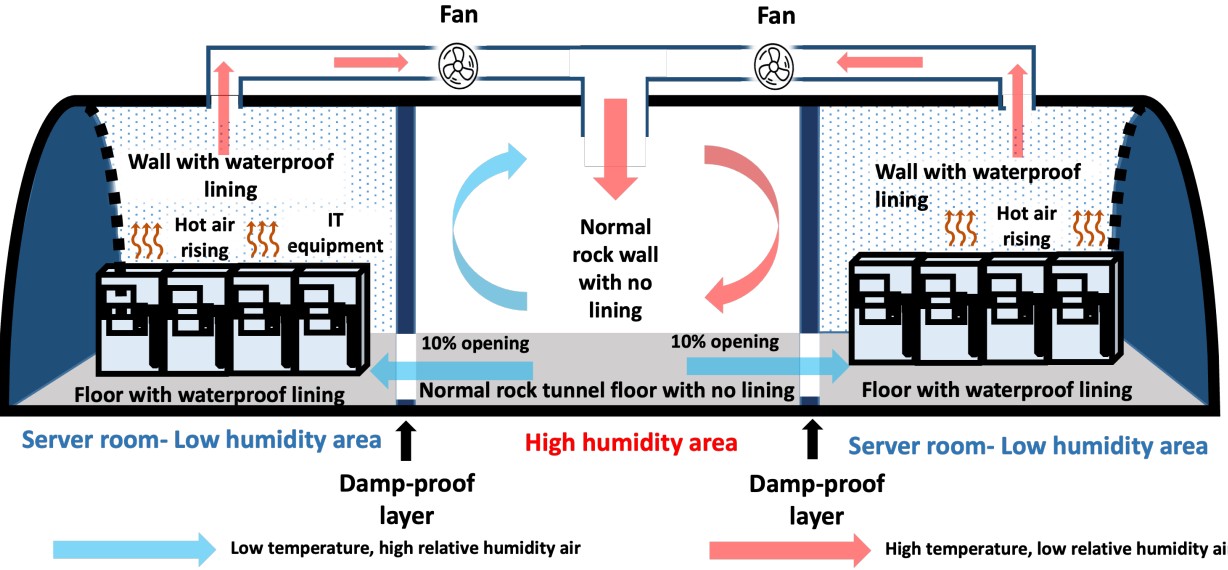

**Figure 17.** The layout of tunnels simulated in scenario 2. The humid section (high-humidity area) is placed between two server rooms (low-humidity area). Ducts are simulated above the server room, channeling high-temperature air from the server rooms into the humid room to be cooled before it re-enters the server rooms via vents at the bottom of the server rooms.

### 3.4.2. Tunnel Design for Scenario 1

Figure 16 illustrates the proposed layout for scenario 1. In this configuration, the left chamber serves as the server room; its tunnel walls and floor are sheathed with a waterproof lining to protect IT equipment from moisture and regulate humidity levels. Conversely, the right chamber is a high-humidity area, deliberately devoid of tunnel walls or floor linings to maximize the utilization of natural cooling properties. An auxiliary exhaust fan, compact in size yet fixed in flow capacity, is positioned at the far end of this setup to instigate airflow and foster a unidirectional current throughout the tunnel. An inter-room aperture, equivalent to 10% of the tunnel's face area (18.8 m$^2$, thereby equating to 1.8 m$^2$), is factored into the simulation to facilitate unrestrained airflow from the server room to the humid area. Given the natural buoyancy of hot air, the ventilation strategy for the server rooms incorporates the installation of hot air outlets at the uppermost point, while cool air inlets are situated at the lowest. In scenarios such as power outages, it is imperative that this opening be sealed to maintain an airtight environment. This precaution is necessary to thwart the attainment of thermal equilibrium between the server and humid rooms' relative humidity levels, safeguarding IT equipment from exposure to conditions surpassing permissible limits.

### 3.4.3. Tunnel Design for Scenario 2

Figure 17 depicts the layout proposed for the second scenario, wherein a humid room is strategically positioned between two server rooms. In this arrangement, the walls and floors of the server rooms are enveloped with waterproof linings, a precautionary measure to shield the IT equipment from potential moisture damage. In contrast, the humid room retains its original state, devoid of any waterproof linings, to exploit the ambient conditions for natural cooling purposes.

### 3.4.4. Moisture and Humidity Control

Although the dry-bulb temperatures align with the ASHRAE Thermal Guidelines from 2014 [7], the tunnel walls at the Osarizawa mine site present considerable dampness, with certain areas experiencing saturation, leaks, and pooling water. Positioning IT equipment within an un-lined tunnel in this environment entails a significant risk of damage. Consequently, it is imperative to install waterproof linings on the server room tunnel walls

and floors to maintain the dryness of the IT equipment. Contrarily, this study advocates for the utilization of humid sections in their existing condition, without any waterproof linings, to leverage the natural cooling capacity inherent in the humid rooms. These specific conditions are emulated by adjusting the rock wetness fractions within the tunnel. The dry rock, analogous to the server room wall, is preset with a value of 0.1, while the saturated rock, representing the humid section, is assigned a value of 1.0, as illustrated in Figure 18.

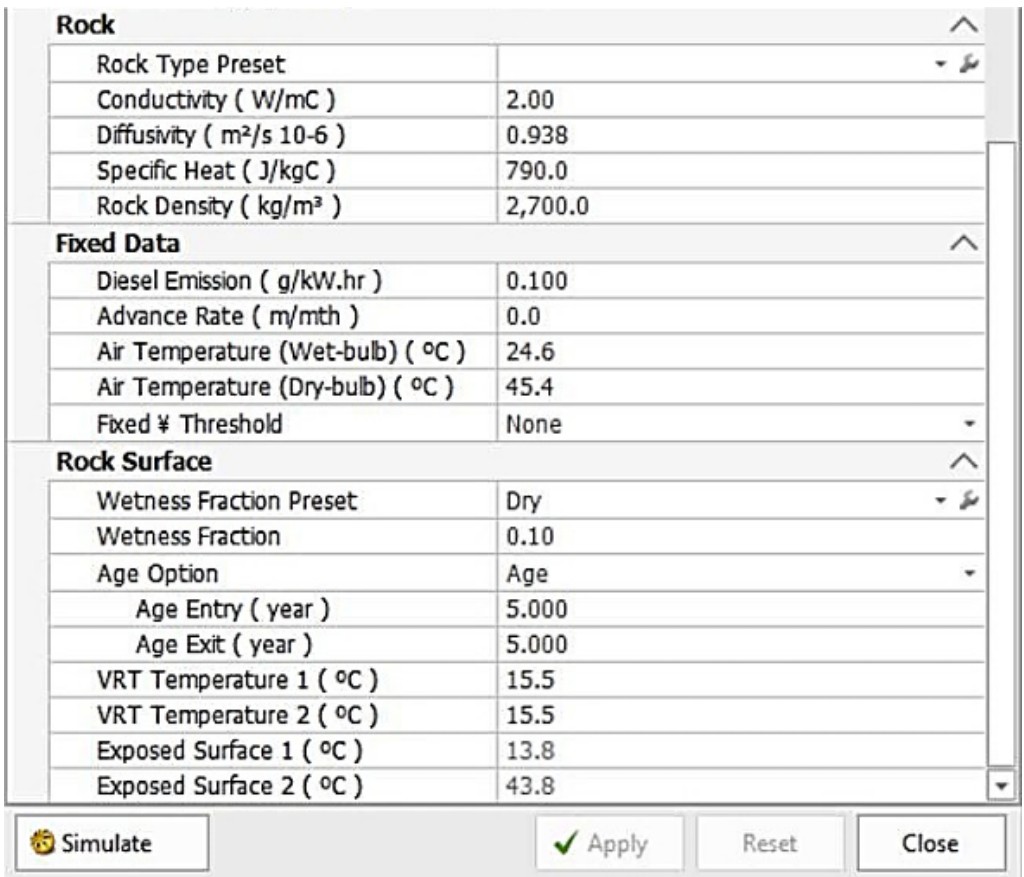

**Figure 18.** Adjusting rock wetness fraction to simulate dry–wet tunnel wall conditions.

### 3.4.5. Input Parameters

The input parameters for the simulation were bifurcated into two categories: fixed environmental parameters, which were predicated on the actual conditions of the Osarizawa mine site, and variable input parameters, which were modulated to discern their impacts on the server room's dry-bulb temperature and humidity. The fixed environmental parameters, preset based on data collated from the Osarizawa mine site, are immutable throughout the simulation. A depiction of the fixed environmental parameter settings is provided in Figure 19. Given the mine's consistent internal temperature of 13 °C year-round and near-saturation humidity levels, both the surface datum dry-bulb and wet-bulb temperatures were equated. Projecting that the data center server room would be situated beyond the tourist-accessible mine segments, a target area approximately 650 m from the entrance was designated as the surface datum reference point. These presets encompass parameters such as the rock wetness fraction, surface datum dry-bulb temperature, and surface datum relative humidity, among others.

| Dynamic | |
|---|---|
| **Environment** | |
| [RESET] | No |
| Air Density Compressible Flow | 1.23 kg/m³ |
| Air Density Incompressible Flow | 1.23 kg/m³ |
| Airway Age | 5.000 year |
| Current Year | 2021.929 |
| Geothermal Gradient | 2.5 C/100m |
| Rock Density | 2,700 kg/m³ |
| Rock Specific Heat | 790.0 J/kgC |
| Rock Thermal Conductivity | 2.00 W/mC |
| Rock Thermal Diffusivity (Readon | 0.938 m²/s 10-6 |
| Rock Wetness Fraction | 0.15 |
| Surface Atmospheric Lapse Rate | 6.4 C/1000m |
| Surface Datum Elevation Above S | -22.3 m |
| Surface Datum of MineGrid | 100.0 m |
| Surface Datum Pressure Barometr | 101.6 kPa |
| Surface Datum Relative Humidity | 100.0 % |
| Surface Datum RockTemp | 13.0 °C |
| Surface Datum Temperature Dry E | 13.0 °C |
| Surface Datum Temperature Wet E | 13.0 °C |
| Surface Temperature Adjust | Yes |

**Figure 19.** Setting fixed environmental parameters based on target area as reference.

The variable input parameters, modulated to evaluate their influence on the dry-bulb temperature and relative humidity of the server rooms, included:

- Varying server room tunnel lengths with a fixed width of 4 m. The tunnel's length was adjusted, serving as a simulation input parameter. For every extension in the server room tunnel length, the total thermal output was increased by 92 kW. Tunnels as long as 1 km were incorporated into the simulations.
- Diverse humid room tunnel lengths, altered to assess the natural cooling capacity corresponding to different volumes. The shift in air temperature is contingent on the tunnel section's internal air volume, while the relative humidity fluctuates based on the air temperature. A voluminous air pocket receiving an equivalent quantum of thermal input would register a marginal escalation in dry-bulb temperature, consequently experiencing a minor uptick in relative humidity.
- Distinct airflow rates, achievable by regulating the fan speeds, thereby varying the airflow volumes permeating the server and humid rooms. Enhanced airflow volumes are capable of dissipating greater quantities of heat and water vapor, thereby amplifying the cooling effect and subsequently impacting the temperature and relative humidity within the server rooms.

## 4. Results

### 4.1. Scenario 1

#### 4.1.1. Different Lengths of Server Room Tunnels

The server room's maximum length was simulated by sequentially increasing the tunnel length in 100 m increments, up to a maximum of 1000 m. Corresponding to each 100 m increase in tunnel length, the data center's thermal output—a critical input parameter in the simulation—was increased by 92 kW. The resulting variations in relative humidity

and dry-bulb temperature of the server rooms at different thermal outputs are cataloged in Figure 20. It should be noted that the exhaust fan's airflow was maintained at a constant 8 m$^3$/s, conforming to the standard airflow specification for a small auxiliary fan within the VentSim software. The simulation outcomes were juxtaposed against the 2011 ASHRAE Thermal Guidelines for data center equipment environmental specifications, specifically class A1's recommended envelope of 17 °C to 28 °C and 60% RH, in addition to the allowable envelope of 20% to 80% relative humidity and 15 °C to 32 °C [7]. The entries in Figure 20 that are congruent with these stipulations are highlighted in green, whereas those that do not satisfy the prerequisites are marked in orange. Specifically, server rooms measuring 200 m and 300 m in length align with the recommended envelope, while those spanning 100 m and 400 m are within the allowable envelope. For server rooms exceeding 500 m in length, the installation of auxiliary cooling systems becomes imperative, as the simulations predict conditions surpassing the maximum allowable dry-bulb temperature.

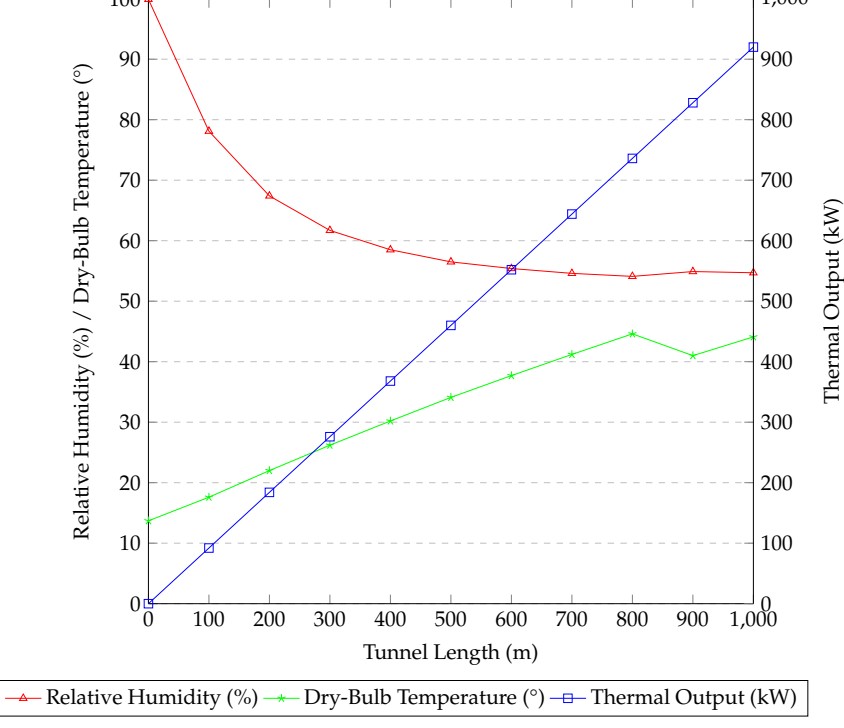

**Figure 20.** The graph demonstrates the correlation between the tunnel length and the corresponding changes in thermal output, relative humidity, and dry-bulb temperature.

### 4.1.2. Scaling up to Two Server Rooms

A simulation was conducted on a series comprising a server room, a humid room, and another server room, as depicted in Figure 21. The server room's length was incrementally increased by 100 m units, ranging from 100 m to 1000 m (SL = 100 m to 1000 m), which also increased its thermal output. The humid room's length remained constant at 100 m (HL = 100 m). Table 8 illustrates the variations in dry-bulb temperature and relative humidity corresponding to the server room's increased thermal output. The cells in green indicate the permissible lengths (and thermal outputs) for the server room, while the ones in orange signify lengths and thermal outputs surpassing the allowable limits. Server rooms with a length of 100 m can be operational without additional cooling systems; however, lengths of 200 m and beyond necessitate supplementary cooling mechanisms.

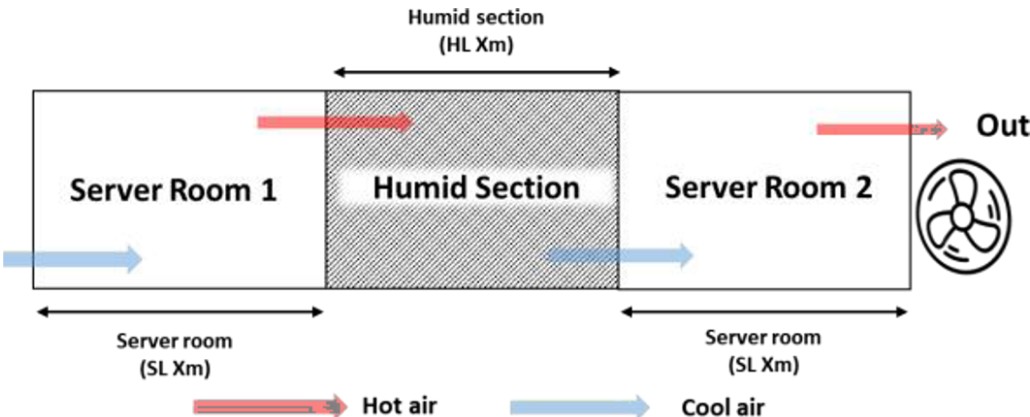

**Figure 21.** Server room–humid room–server room arrangement; length of server room (SL) is adjusted.

**Table 8.** Simulation results for various server room lengths in the server room–humid room–server room configuration.

| Tunnel (m) | Server Room | Relative Humidity (%) | Dry-Bulb Temp. (°) |
|---|---|---|---|
| 0 (Initial) | 1 | 99.9 | 13.7 |
| | 2 | 99.9 | 13.7 |
| 100 | 1 | 78.0 | 18.4 |
| | 2 | 56.9 | 24.9 |
| 200 | 1 | 67.3 | 22.9 |
| | 2 | 38.9 | 34.7 |
| 300 | 1 | 61.7 | 27.1 |
| | 2 | 29.6 | 43.2 |
| 400 | 1 | 58.4 | 31.2 |
| | 2 | 24.1 | 50.8 |
| 500 | 1 | 56.5 | 35.1 |
| | 2 | 20.6 | 57.4 |

### 4.1.3. Simulating the Cooling Capacity of Different Humid Room Tunnel Lengths

The impact of variations in the humid section length (HL ranging from 100 m to 1500 m) on the relative humidity and dry-bulb temperature was simulated, with the server room length fixed at 200 m (SL = 200 m) and a constant airflow of 8 $m^3/s$. Table 9 illustrates the simulation outcomes, emphasizing the dry-bulb temperatures in server room 2. Temperatures within the permissible range are highlighted in green, whereas those failing to meet the environmental standards are marked in orange. For server rooms with a length of 200 m (SL200), an intermediate humid room of at least 300 m is necessary.

Figure 22 depicts the correlation between the extension of the humid room and the subsequent reduction in dry-bulb temperature. The segment highlighted in green (15° to 32°) represents the temperature spectrum compliant with data center standards. For two 200 m server rooms in a server room–humid room–server room configuration, elongating the humid room from 100 m to 300 m effectively reduces the dry-bulb temperature within acceptable limits. Nevertheless, the cooling capacity is insufficient for server room 2 in the SL = 300 m, 400 m, and 500 m arrangements.

The elongation of the humid room consistently affects all server room lengths (SL100 to SL500), diminishing the dry-bulb temperature and elevating the relative humidity. For SL100, a change in DBT of less than 0.5° begins at 500 m. This threshold is reached at 600 m for SL200 and SL300, and at 700 m for SL400 and SL500. Across all server room dimensions, variations in the dry-bulb temperature are negligible (≤0.2°) beyond 900 m, indicating the minimal impact of further extending the humid room tunnel. In the absence of additional cooling mechanisms, a substantial extension of the humid room tunnel is essential. A

longer humid room correlates with increased relative humidity in server room 2. For every server room category, a change in relative humidity of less than 0.5% (RH ≤ 0.5) is observed beyond 900 m. Although the relative humidity across all server room lengths adheres to the prescribed guidelines, extending the humid room can elevate the RH from the allowable to the recommended range, mitigating the risk of electrostatic discharge (ESD). Figure 23 demonstrates the association between the humid room's length and the server rooms' relative humidity. The green zone in the diagram indicates the acceptable relative humidity levels; humid rooms ranging from 100 m to 500 m all satisfy the data center's humidity prerequisites. Extending the humid room past 900 m yields no significant rise in relative humidity or reduction in dry-bulb temperatures (both change at rates of <0.5%, <0.5°), attributable to the near-equilibrium state of the temperature and relative humidity gradient between the server and humid rooms.

**Table 9.** Changes in relative humidity and dry-bulb temperature in server room 2 due to variations in humid room length.

| Humid Room Length (m) | Relative Humidity (%) | Dry-Bulb Temp. (°) |
|---|---|---|
| 100 | 38.9 | 34.7 |
| 200 | 49.2 | 32.1 |
| 300 | 56.2 | 30.6 |
| 400 | 61.1 | 29.7 |
| 500 | 64.1 | 29.1 |
| 600 | 65.9 | 28.7 |
| 700 | 67.0 | 28.5 |
| 800 | 67.1 | 28.3 |
| 900 | 68.1 | 28.2 |
| 1000 | 68.3 | 28.1 |
| 1100 | 68.4 | 28.0 |
| 1200 | 68.5 | 28.0 |
| 1300 | 68.6 | 27.9 |
| 1400 | 68.6 | 27.8 |
| 1500 | 68.6 | 27.8 |

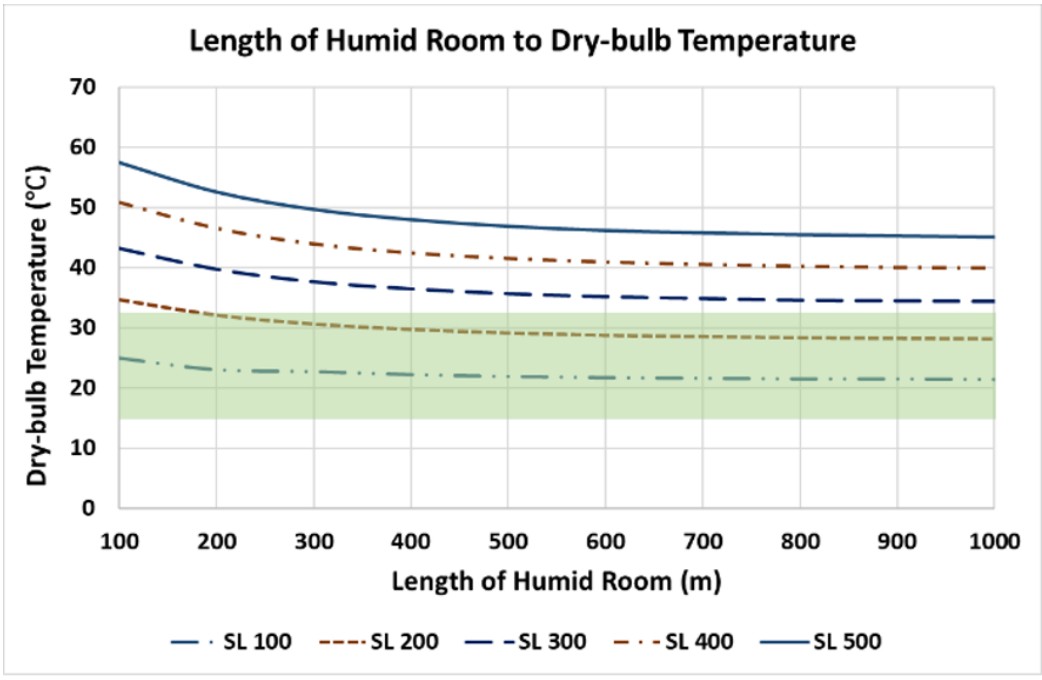

**Figure 22.** The decrease in dry-bulb temperature of server room 2 due to increase in length of humid tunnel, allowable envelope (light green).

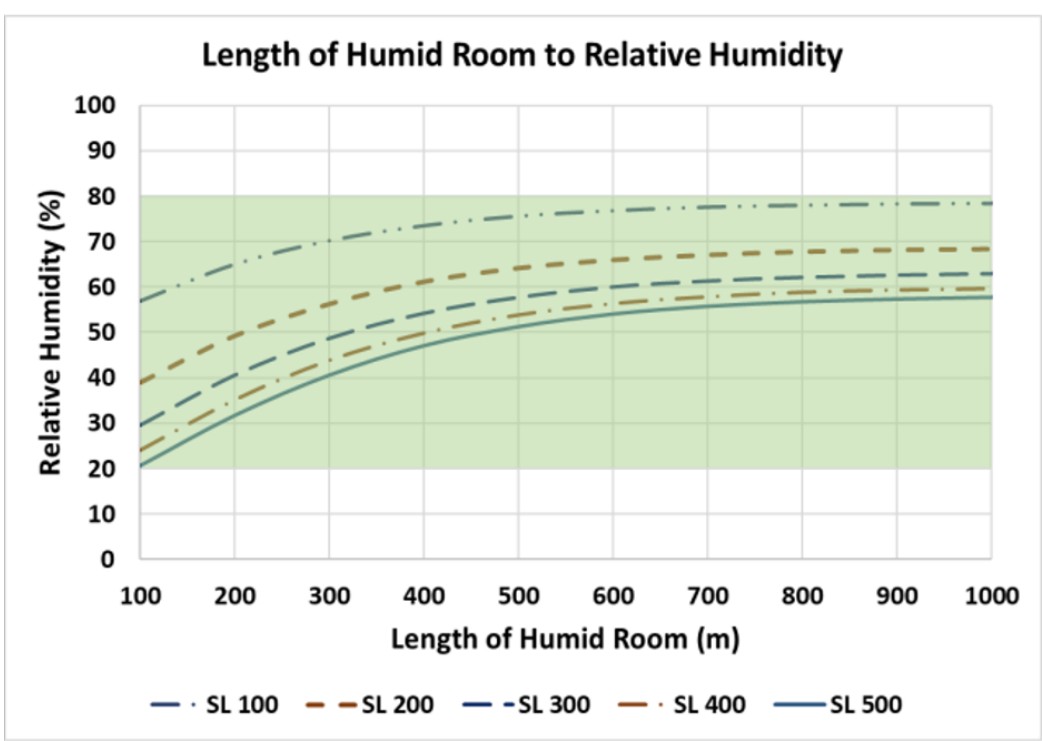

**Figure 23.** The increase in relative humidity of server room 2 due to increase in length of humid tunnel, allowable envelope (light green).

### 4.1.4. Different Airflows

The configurations of the auxiliary exhaust fans, situated at the termination of the layout, were modified to simulate varying airflows across five server rooms structured in a server room–humid room–server room sequence. Operating expenses are predominantly affected by the energy consumption of these supplementary fans and escalate in direct correlation with increased airflow. Airflows ranging from 5 m$^3$/s to 9 m$^3$/s proved inadequate in terms of ventilation and cooling efficacy, with server rooms 4 and 5 surpassing the advised thresholds for dry-bulb temperatures. Conversely, airflows extending to 10 m$^3$/s or more were deemed excessive, as evidenced by the suboptimal dry-bulb temperature and increased relative humidity in server room 1, deviating from the recommended parameters. A pragmatic approach involves the implementation of a 7 m$^3$/s airflow, complemented by supplementary cooling mechanisms pre-conditioning the air prior to its ingress into server rooms 4 and 5. Table 10 delineates the variance in dry-bulb temperatures and relative humidity observed across the five server rooms subject to differing airflows.

**Table 10.** Dry-bulb temperature and relative humidity in different server rooms at various airflows.

| Airflow (m$^3$/s) | 5 | 7 | 8 | 9 | 10 | 12 | 15 |
|---|---|---|---|---|---|---|---|
| Room | DBT (°C)/RH (%) | DBT (°C)/RH (%) | DBT (°C)/RH (%) | DBT (°C)/RH (%) | DBT (°C)/RH (%) | DBT (°C)/RH (%) | DBT (°C)/RH (%) |
| 1 | 21.2/70.7 | 18.2/76.1 | 18.2/78.2 | 17.1/79.7 | 17.5/81.1 | 16.0/83.4 | 16.2/85.7 |
| 2 | 31.1/44.4 | 25.7/52.3 | 25.7/56.1 | 23.0/58.8 | 22.7/62.3 | 20.5/65.4 | 19.7/70.3 |
| 3 | 37.7/36.9 | 31.0/43.8 | 31.0/47.7 | 27.3/50.5 | 26.5/55.0 | 23.9/57.5 | 22.2/63.6 |
| 4 | 42.3/34.7 | 34.9/40.5 | 34.9/44.3 | 30.6/47.0 | 29.4/51.9 | 26.4/53.8 | 24.2/60.4 |
| 5 | 45.7/34.3 | 38.1/39.7 | 38.1/43.2 | 33.3/46.0 | 31.6/50.8 | 28.6/52.5 | 25.7/58.8 |
| Power (kW) | 10.7 | 26.8 | 26.8 | 57.0 | 86.0 | 135.1 | 290.2 |
| Cost (USD) | 9416 | 23,466 | 23,466 | 49,916 | 75,327 | 118,320 | 254,227 |

### 4.1.5. Different Server Room Arrangement

Different server room lengths were arranged in various sequences, with a humid room consistently positioned between two server rooms. Various airflows were simulated to determine the minimum airflow necessary to achieve an allowable envelope. VentSim automatically calculates operating costs based on the operational power required by the ventilation equipment. Table 11 presents the results of different server room length arrangements, along with their respective power consumption and operational costs. Green cells indicate acceptable conditions, while orange cells denote conditions that fail to meet data center standards. A 400 m server room is the most energy and cost efficient. Although the required airflow is larger at 8 m$^3$/s compared to the minimum of 7 m$^3$/s, the absence of resistance and friction/shock losses in this configuration results in reduced power consumption. When the minimum required airflow is 10 m$^3$/s, power consumption and operational costs increase; however, the server room reaches allowable envelope conditions more quickly due to higher air exchange rates. VentSim simulations were performed to test the maximum server room length, the natural cooling provided by humid sections, and various server room–humid room configurations. For a single server room, the maximum length (with proportional thermal input) is 400 m. At 500 m (460 kW thermal output), the dry-bulb temperature exceeds the allowable envelope.

**Table 11.** Airflow, power consumption, and cost of different server room–humid room arrangements.

| Server Room Length (m) | Room Length Ratio | Airflow Required (m$^3$/s) | Power (kW) | Operating Cost (USD) |
|---|---|---|---|---|
| 100 | 1:1:1:1:1 | 8 | 28 | 24,546 |
| | 2:1:2 | 10 | 39.1 | 34,255 |
| 200 | 2:2:1 | 7 | 13.4 | 11,750 |
| | 1:2:2 | 7 | 13.4 | 11,750 |
| 300 | 1:1:3 | 9 | 28.5 | 24,972 |
| | 3:1:1 | 10 | 39.1 | 34,255 |
| 400 | 4:0 | 8 | 12 | 10,530 |
| 500 | 5:0 | 10 | 28.5 | 20,570 |

In scenarios with two server rooms in a server room–humid room–server room configuration, only 100 m (92 kW thermal output) can be installed consecutively without supplemental cooling systems. Various lengths of humid rooms were simulated to study the natural cooling capacity of the humid rooms. Maximum cooling occurs at 900 m; beyond 900 m, the increase in relative humidity and the decrease in dry-bulb temperature are less than 0.5% and 0.5 °C, respectively. This correlates with the absence of a steep thermal gradient and indicates that the outlet air from the server room and the air in the moist room are nearly at thermal and moisture equilibrium. These simulations confirm that by not lining the humid room walls with waterproof linings, it is possible to enhance the relative humidity in the server room outlet area and reduce the dry-bulb temperature before it enters the subsequent server room.

### 4.2. Scenario 2

Two server rooms, each with a fixed length of 100 m, and humid rooms with variable lengths were examined in the simulation, as depicted in Figure 17. The server rooms were set at a standard length (SL) of 100 m, while the humid room length (HL) was varied from 100 m to 500 m and extended up to 1000 m. The dry-bulb temperatures, humidity levels, and airflows were recorded.

Figure 24 illustrates the impact on the dry-bulb temperature in the second server room (S2) attributable to modifications in both airflow and humid room length. The most substantial temperature reduction occurs when the humid room length is extended from 100 m to 200 m, with the dry-bulb temperature (DBT) dropping from approximately 80 °C to around 60 °C. A further extension to 300 m decreases the temperature only marginally to approximately 50 °C. Beyond this point, additional increases in the humid room's length

do not significantly affect the DBT, and none of the configurations meet the permissible temperature conditions for the data server rooms.

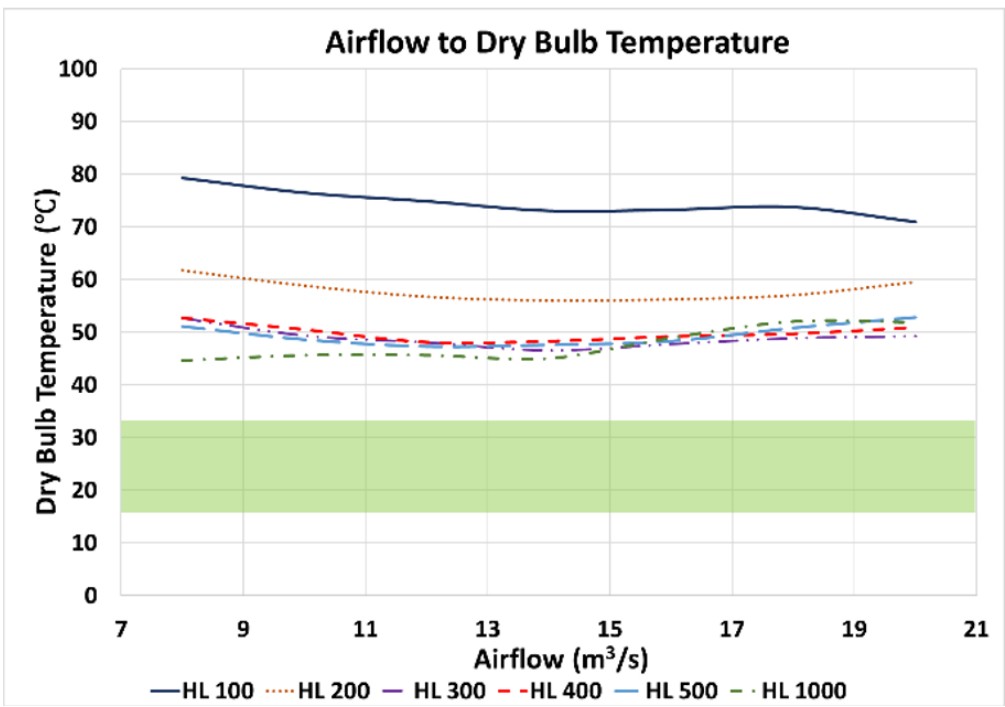

**Figure 24.** Changes in dry-bulb temperature with increases in airflow and humid room length, allowable envelope (light green).

Figure 25 delineates the alterations in the relative humidity within server room 2 caused by changes in both airflow and humid room length. A consistent 9–10% increase in relative humidity is observed as airflow intensifies from 8 m³/s to 20 m³/s across all humid room lengths. Specifically, for humid rooms measuring 400 m, airflow exceeding 13 m³/s is deemed excessive. Extending the humid room to a length of 1000 m inadvertently propels the relative humidity beyond the acceptable limits. These findings indicate the insufficiency of natural cooling provided by this configuration for server rooms, even at the minimal length of 100 m.

Subsequent simulation outcomes, as shown in Figure 25, suggest a potential drawback; indefinitely extending the humid room's length eventually results in the server room's relative humidity surpassing acceptable levels. Increasing the airflow effectively reduces the dry-bulb temperature but only to a certain extent. Figures 26–28 present simulation outcomes underscoring that for all considered humid room lengths, enhancing the exhaust fans' airflow in ducts up to 14 m³/s is beneficial in lowering temperatures. Any further increase reverses this trend, elevating the dry-bulb temperature instead. This pattern underscores an optimal airflow of 14 m³/s for this particular ventilation arrangement. The respective figures show the correlation between increasing airflow and dry-bulb temperature for humid rooms of lengths 100 m, 500 m, and 1000 m.

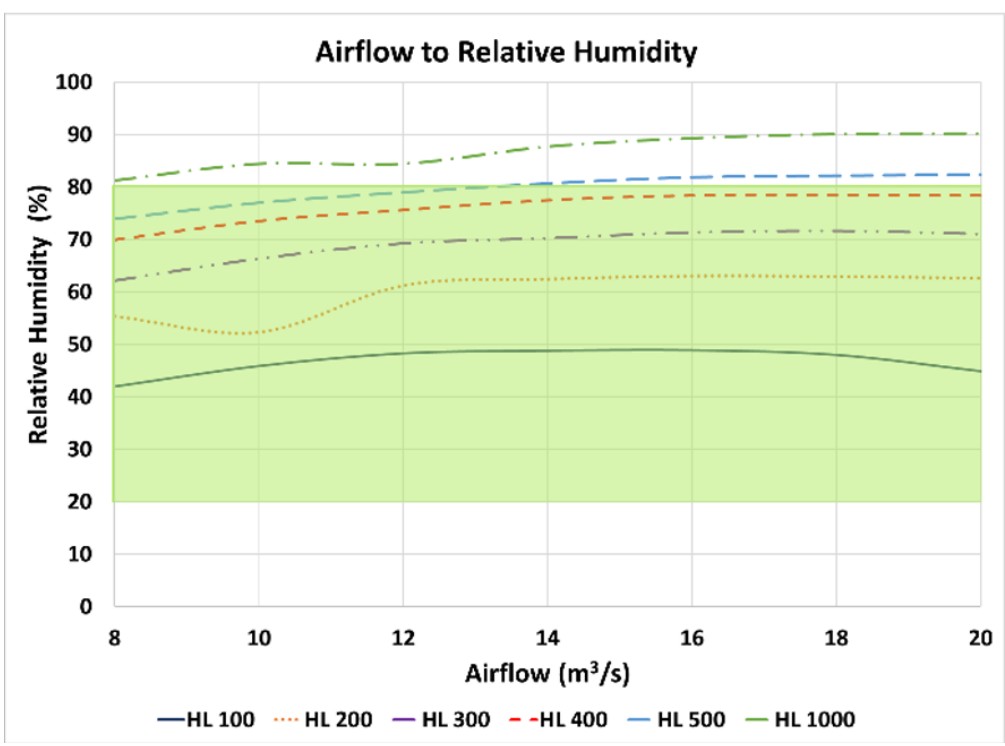

**Figure 25.** Changes in relative humidity with increases in airflow and humid room length, allowable envelope (light green).

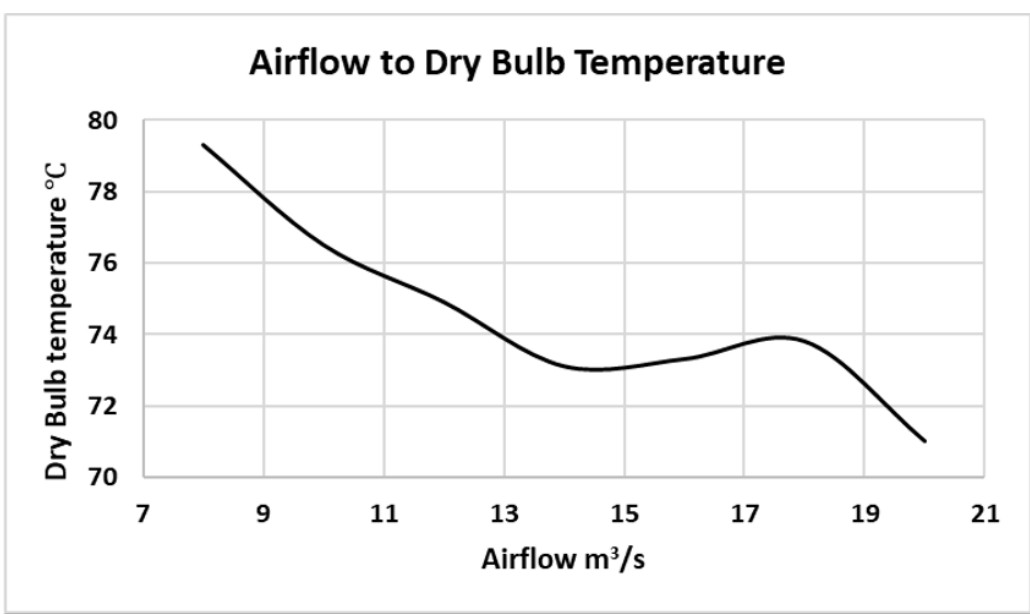

**Figure 26.** Changes in dry-bulb temperature of server room 2 at SL = 100 m with increase in airflow.

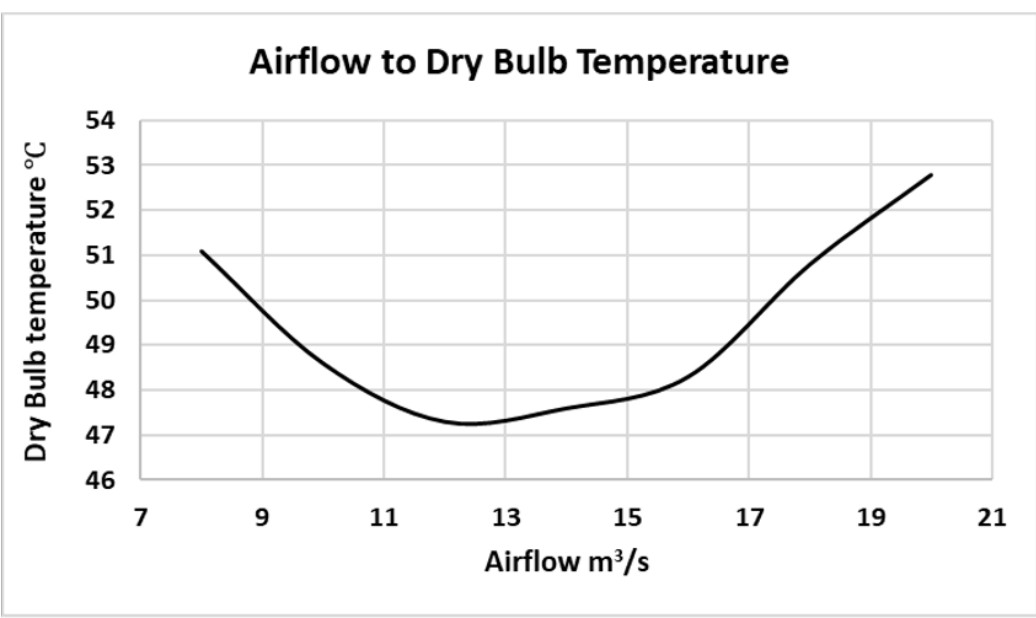

**Figure 27.** Changes in dry-bulb temperature of server room 2 at SL = 500 m with increase in airflow.

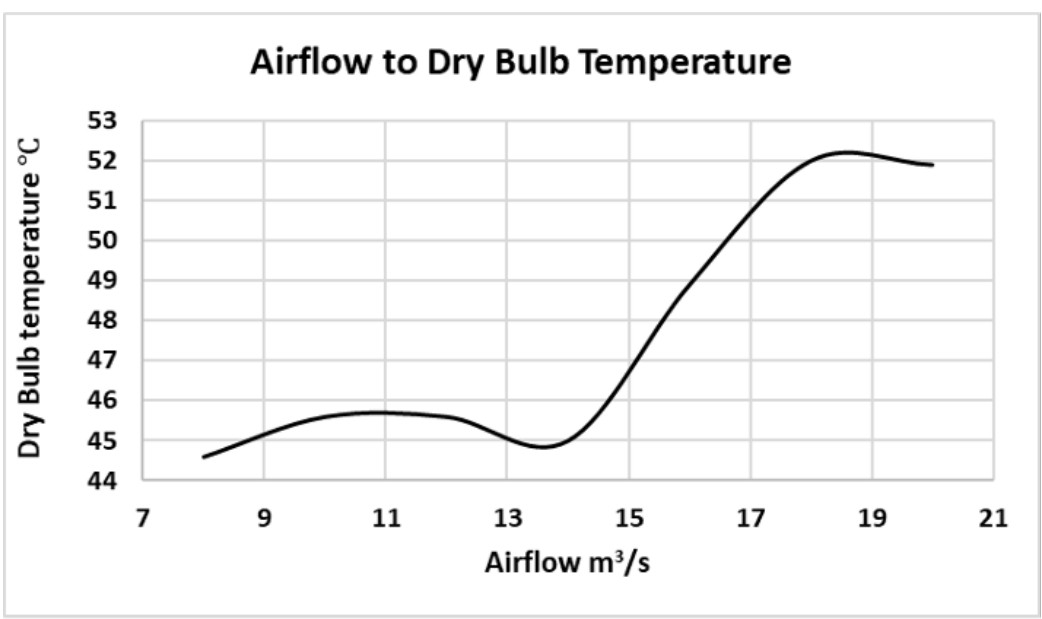

**Figure 28.** Changes in dry-bulb temperature of server room 2 at SL = 1000 m with increase in airflow.

*4.3. Visualizing Simulation Results Using Point Cloud*

Simulation outcomes were superimposed on the "dense" point cloud derived using photogrammetry techniques. The temperature gradient spanning the humid room, server room, and outlet was extracted from the VentSim simulations, and the point cloud's color scheme was manipulated to enhance the visual representation of these results. The simulations were conducted based on the layout specified in scenario 1.

Figure 29 displays a screenshot of the simulated tunnel, serving as a benchmark for modifying the point cloud's color hues to depict the thermal gradient, which extends from lower temperatures (13.7 °C) to higher ones (30.6 °C). Subsequently, Figures 30 and 31 present a point cloud that characterizes this thermal gradient. The constructed point cloud of the tunnel spans a length of 10 m and was magnified 30-fold to simulate the conditions of a 300 m tunnel accurately.

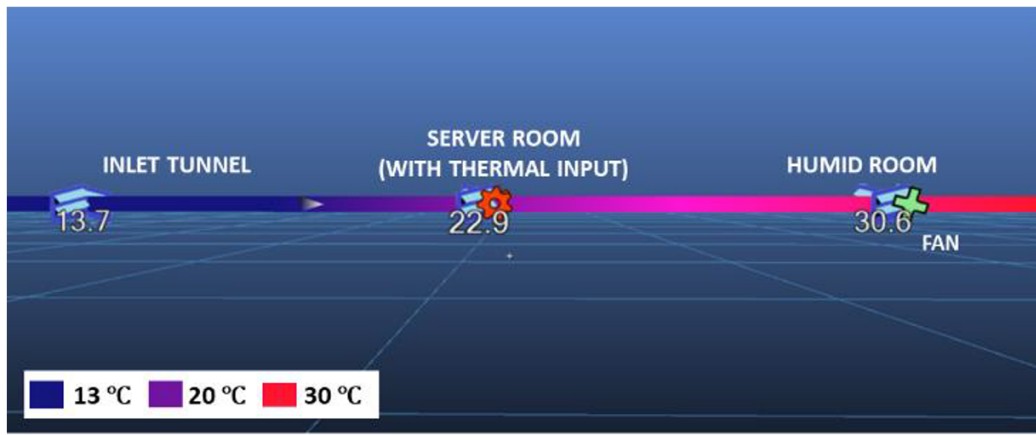

**Figure 29.** Point cloud of tunnel illustrating thermal gradient.

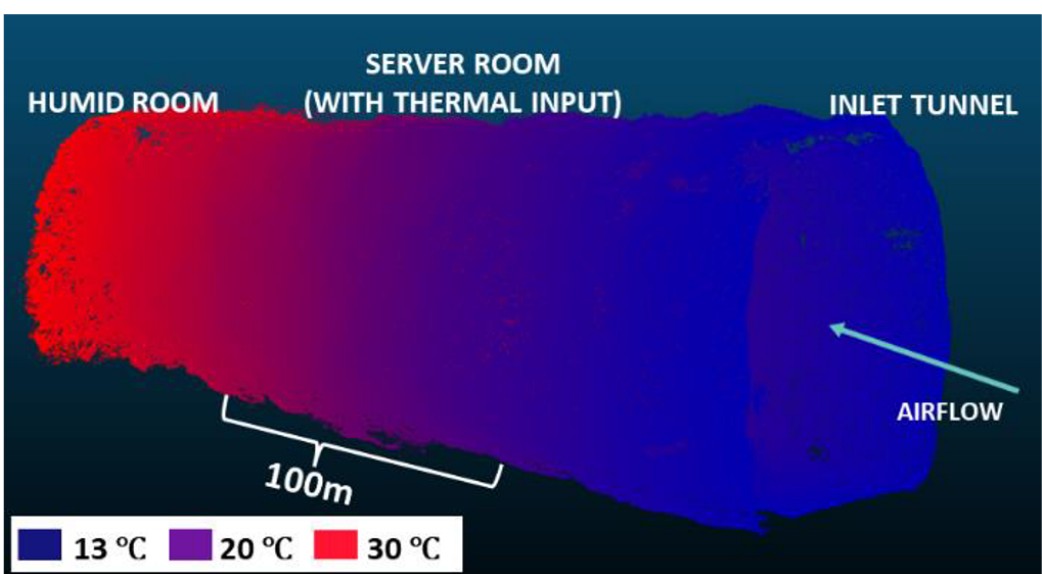

**Figure 30.** Point cloud of tunnel illustrating thermal gradient (flat view).

### 4.4. Summary of Results

Arched tunnels, each 4 m in width and 5 m in height, were simulated under two distinct scenarios: scenario 1, depicted in Figure 16, and scenario 2, illustrated in Figure 17. The simulations involved varying the rock wetness fraction to emulate both dry, waterproof-lined server room walls and the naturally moist walls of the humid rooms. Diverse configurations, dimensions of server and humid rooms, and airflow conditions were explored.

In scenario 1, a linear airflow pattern was employed, with an exhaust fan positioned at the arrangement's terminus, directing airflow through a server room–humid room–server room sequence. This setup revealed that a server room extending 400 m was notably more energy and cost efficient relative to alternative configurations. Scenario 2, conversely, was assessed by installing air ducts atop the server rooms, through which high-temperature air was propelled into the humid room by exhaust fans within the ducts. For the shortest server room tunnel examined (100 m), this design failed to supply adequate natural cooling to satisfy data center temperature norms. Elongating the humid room caused an exceedance of the requisite relative humidity levels, while increasing the airflow beyond an optimal threshold inadvertently provoked a temperature surge. This phenomenon aligns with the operational paradigm of a fan's best efficiency point (BEP), where the fan functions most cost-effectively concerning both energy use and maintenance [54]. When a fan's airflow surpasses its BEP, it demands additional electrical power for the same output. Given that the heat produced by fans equals the input electrical power [52] and is imparted to the

airflow, increased power expenditure translates to heightened thermal output. Notably, VentSim incorporates fan-generated heat (including fixed flow) within its thermal models, precluding the need for separate fan-based heat sources [17].

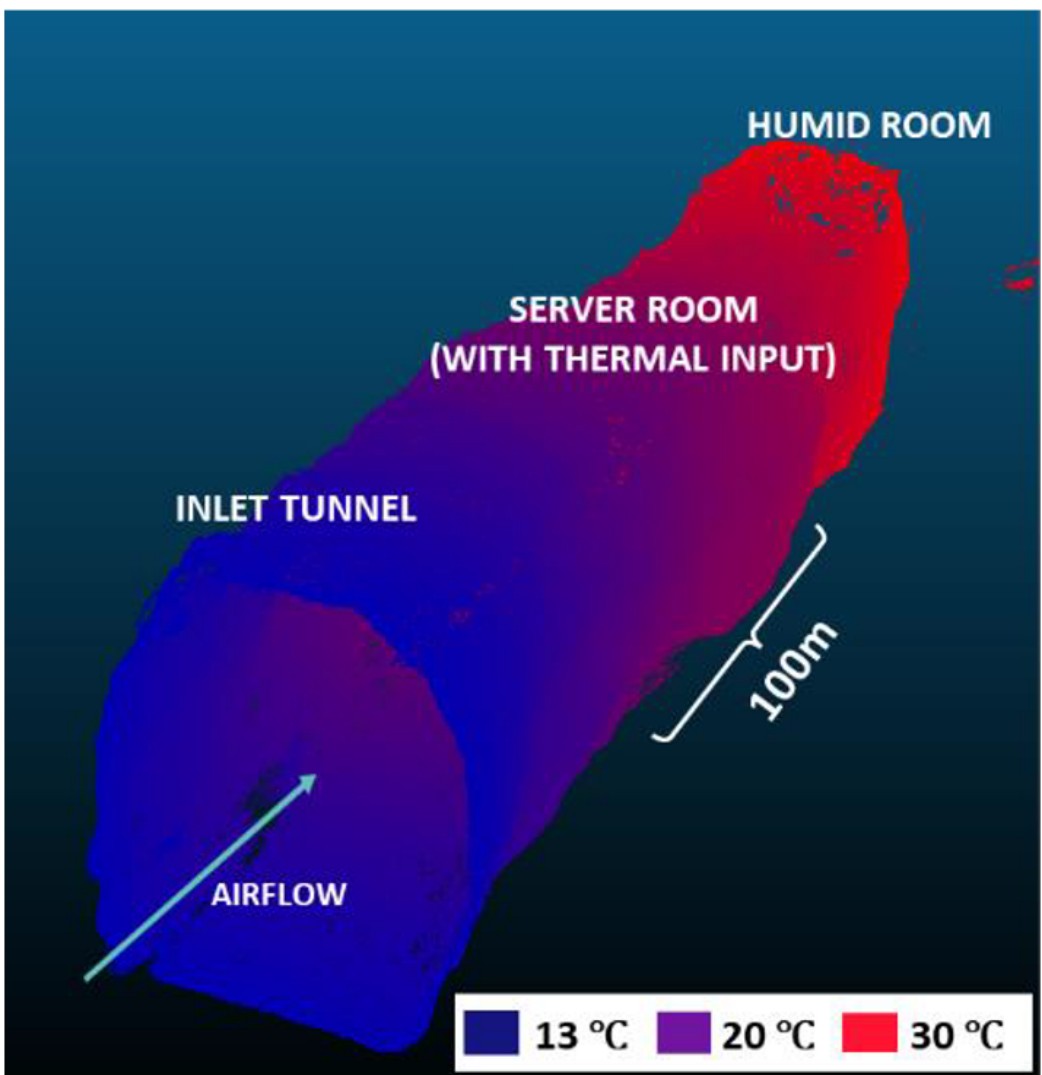

**Figure 31.** Point cloud of tunnel illustrating thermal gradient (another angle view).

Comparative analysis of the simulations underscores scenario 1's superiority over scenario 2 in achieving sufficient natural cooling for a data center, evidenced by its proficiency in keeping both the dry-bulb temperature and relative humidity within permissible data center parameters. For scenario 2, supplementary cooling mechanisms would be indispensable to meet the stringent temperature criteria. Furthermore, dehumidification systems would be requisite to uphold acceptable relative humidity levels. Conversely, scenario 1 poses a heightened risk in instances of power disruptions or ventilation equipment malfunctions. Due to its linear airflow architecture, the entire arrangement's ventilation hinges on a single exhaust fan; a malfunction in this unit would compromise the ventilation in all sequential server rooms.

In contrast, scenario 2 allocates an individual duct and exhaust fan to each server room, localizing the impact of potential fan failures. Consequently, only the server room linked directly to the compromised fan would suffer ventilation disruption, rendering scenario 2 inherently less risky and mitigating potential losses ensuing from ventilation breakdowns.

## 5. Discussion

This study embarked on the ambitious task of creating a digital twin of a tunnel within the Osarizawa mine site using photogrammetry, intending to leverage this twin as a visual aid for representing complex simulation outcomes. The primary objective was to scrutinize the environmental preconditions of the Osarizawa mine site, probing its suitability for accommodating a data center. Although the intricacies of integrating the digital twin with VentSim presented insurmountable challenges, circumventing this by employing a simplified model on VentSim for the simulations was possible. These simulations were pivotal in assessing the technical feasibility of transmuting the Osarizawa mine site into a data center hub, requiring meticulous manipulation of input parameters—spanning the lengths of server and humid rooms, thermal output, and induced airflow—and close observation of the resultant variations in the server room's dry-bulb temperature and relative humidity.

The simulation insights suggest that, contingent upon the installation of waterproof linings along the tunnel walls and floors within the server room, the underground expanse of the Osarizawa mine is a viable candidate for data center conversion. These conclusions stem from extensive simulations of linear induced airflow, the calibration of simulation parameters, and the determination of optimal airflow, humid room dimensions, and maximal server room extension.

Moreover, the humid rooms, initially conceptualized as vacant spaces nestled between server rooms, possess the potential for alternative applications. The consistent low temperatures historically characteristic of the Osarizawa mine's subterranean tunnels have catered to their use as wine storage and fermentation sites. However, the thermal alterations consequent to the installation of data servers would compromise these conditions, necessitating a shift in utility. Agricultural pursuits, particularly fungiculture, emerge as viable alternatives, thriving in the 18 °C to 26 °C range fostered by hydroponic methodologies and energy-efficient LED lighting. Such ventures would demand significantly less energy compared to the operation of a data center. The requisite sterility for fungiculture could be reliably maintained in the controlled environments of underground humid rooms. Figure 32 depicts a successful model of mushroom cultivation within subterranean confines on the periphery of Paris. Furthermore, excessive heat outputs, particularly those surpassing 30 °C, could be harnessed, stored, and redistributed to proximal structures, an approach that garners detailed exploration within Paludetto et al.'s study [47].

Despite the promising outcomes, the scope of this research was confined to the preliminary assessment of environmental conditions against the ASHRAE 2011 Thermal Guidelines [7], not extending to the intricate considerations of equipment layout, such as hot and cold aisles, elevated flooring, electrostatic discharge (ESD)-preventative flooring, or advanced cooling techniques. Additionally, the simulations did not account for the cumulative thermal output following the expansion of server room tunnels beyond 500 m or the integration of multiple units. Future initiatives could incorporate comprehensive data of the actual mine layout to optimize the existing ventilation structures. For instance, exhaust fans might be linked to an existing shaft to facilitate heat dissipation through natural convection.

This study's digital twin generation was limited to a 10 m tunnel segment, serving purely for visualization. Comprehensive modeling, however, would necessitate an extensive digital twin, encapsulating the full length of the tunnels under consideration. The incorporation of smart sensors within the digital twin could pioneer real-time monitoring capabilities, laying the groundwork for a ventilation-on-demand (VOD) system. Such an advanced setup would autonomously uphold optimal conditions for IT equipment, governed by real-time data. Envisioning further, establishing a responsive loop between sensors and ventilation apparatus could enable dynamic adjustments, including the activation/deactivation of fans, valves, and other cooling systems in direct response to sensor feedback.

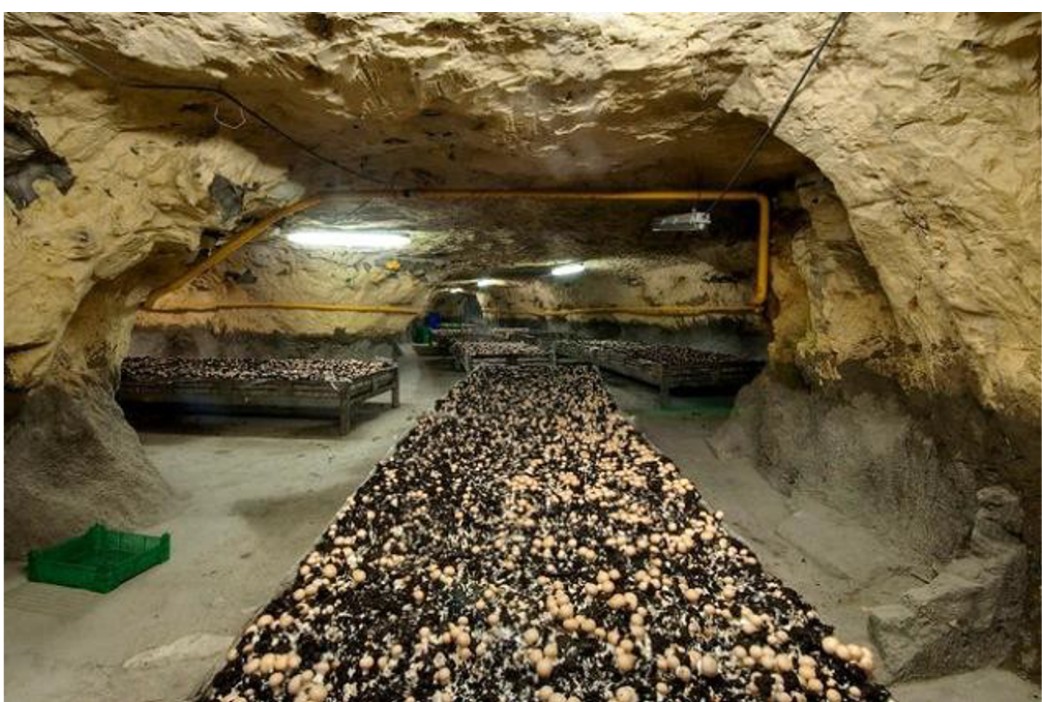

**Figure 32.** Button mushrooms grow underground in Montesson, outside Paris—The Local.

## 6. Conclusions

This research represents a pioneering effort in developing a digital twin of a tunnel within the Osarizawa mine through photogrammetry, aimed at utilizing this twin for simulations to assess the site's environmental conditions and its suitability for hosting a data center. The successful creation of the digital twin, albeit with limitations in modifying the model's parameters in VentSim, led to the development of a simplified model, essential for these simulations. These simulations were instrumental in evaluating the technical feasibility of transforming the Osarizawa mine's subterranean spaces into a data center, analyzing various input parameters and their effects on the server room's temperature and humidity.

Our findings indicate the technical plausibility of re-purposing the underground space for data center purposes, especially when considering a waterproof lining for the server room. The simulations helped identify the minimal required airflow, the ideal humid room length, and the maximum server room length, enhancing our understanding of the site's potential. Furthermore, this research opens new possibilities for alternative uses of the humid rooms, such as for agricultural or fungicultural purposes, leveraging hydroponic and LED technologies. This controlled environment, exemplified by successful mushroom cultivation in similar settings, could significantly contribute to sustainable underground farming practices.

Additionally, the study touches on the concept of reusing excess heat for nearby structures as a sustainable solution when outlet temperatures exceed optimal ranges. However, our study leaves several areas for future exploration, including equipment placement within server rooms and comprehensive modeling of the entire mine. The integration of smart technologies, potentially leading to real-time monitoring and VOD systems, represents an exciting avenue for future research. Such systems could autonomously optimize conditions for IT equipment, adjusting cooling systems based on real-time data.

Crucially, this study highlights the need for a more in-depth analysis of the long-term environmental and sustainability impacts of such transformations. Re-purposing mines like Osarizawa not only offers technical feasibility for data center operations but also presents an opportunity to re-examine and mitigate the broader environmental impacts of these activities. Our research underscores the importance of considering these aspects in

the Abstract and Conclusions, to emphasize the sustainability considerations inherent in such projects.

In conclusion, this study not only confirms the technical feasibility of re-purposing the Osarizawa mine for data center operations, but also sheds light on promising alternative uses and significant opportunities for technological advancements in sustainable underground space utilization.

**Author Contributions:** Conceptualization, H.I., N.E.B.M. and T.A.; methodology, H.I., M.A.M. and Y.K.; software, H.I. and H.T.; validation, H.I., M.A.M. and B.B.S.; formal analysis, H.I. and H.T.; investigation, H.I., N.E.B.M. and B.B.S.; resources, H.I.; data curation, H.I.; writing—original draft preparation, H.I.; writing—review and editing, H.I., M.A.M. and Y.K.; visualization, H.I.; supervision, M.A.M. and Y.K.; project administration, M.A.M. and Y.K.; funding acquisition, N.E.B.M. and T.A. All authors have read and agreed to the published version of the manuscript.

**Funding:** This research received no external funding.

**Institutional Review Board Statement:** Not applicable.

**Informed Consent Statement:** Not applicable.

**Data Availability Statement:** No new data were created in this study

**Conflicts of Interest:** The authors declare no conflict of interest. The funders had no role in the design of the study; in the collection, analyses, or interpretation of data; in the writing of the manuscript; or in the decision to publish the results.

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
