# Peer review of "Digital Twin Technology in Data Center Simulations: Evaluating the Feasibility of a Former Mine Site"

_sustainability, doi:10.3390/su152316176_

Round 1

Reviewer 1 Report

Comments and Suggestions for Authors

The study provides a comprehensive overview of a groundbreaking study that proposes the repurposing of the disused Osarizawa mine in Akita, Japan, to establish a sustainable data center. The innovation lies in leveraging the consistently low tunnel temperatures of the mine to offset reclamation costs and provide a new economic avenue. The study assesses the feasibility of this transformation through the collection of environmental data and meticulous ventilation simulations. The study showcases an innovative approach to addressing the provisional status of mine sites by proposing a practical and sustainable solution for reclamation. The research's credibility is bolstered by its commitment to gathering comprehensive environmental data from the Osarizawa mine site, demonstrating a thorough understanding of the project's context. The ventilation simulations explore a wide range of scenarios, including diverse ventilation configurations, data server room dimensions, thermal outputs, and humid room cooling capabilities. This approach adds depth to the study and increases its applicability to real-world scenarios.

Suggestions for Minor Corrections:

Clarity in Ventilation Simulations: While the study mentions the meticulous ventilation simulations, it would enhance clarity to briefly outline the key findings or insights gained from these simulations. This will provide readers with a more concrete understanding of the study's outcomes.

Specify Environmental Data: Consider briefly specifying the types of environmental data collected from the Osarizawa mine site. This additional detail can enhance the transparency of the study and allow readers to better grasp the basis for the proposed data center establishment.

Quantify Economic Benefits: To strengthen the economic argument for repurposing the mine site, consider including a brief discussion or quantification of the anticipated economic benefits. This could include cost savings, revenue generation, or other relevant financial considerations.

References: double check

Author Response

Dear Reviewer,

Thank you for your insightful comments on our manuscript. We are grateful for the opportunity to refine our work based on your feedback and are excited to apply these learnings to future research. Below, we outline how we have addressed each point and our approach for future studies:

Clarity in Ventilation Simulations: We acknowledge the need for clearer explanations of our ventilation simulations. Accordingly, we have revised this section for improved clarity, ensuring that our methodologies and findings are easily understandable. This refinement will enhance the manuscript's value to readers and fellow researchers.

Specify Environmental Data: We agree with the importance of specifying the environmental data collected, especially given its critical role in our study. In the revised manuscript, we have emphasized the significance of the onsite measured data, such as humidity and temperature, to our research. This detail will provide readers with a deeper understanding of the environmental considerations pivotal to our data center proposal.

Quantify Economic Benefits: The suggestion to quantify economic benefits has sparked a new research idea for our team. While we have utilized power cost trends in our current simulations, we acknowledge that a comprehensive economic evaluation, including factors beyond power costs, is essential. We plan to explore this in future research, expanding the scope and depth of our economic analysis.

References: We are committed to ensuring the accuracy and relevance of our references. We will thoroughly review and adjust them to meet the journal's standards, including a careful reassessment of self-citations.

We appreciate the opportunity to respond to your feedback and believe that these adjustments significantly improve our manuscript. Moreover, your suggestions have provided valuable guidance for our future research directions, helping us to prioritize areas of exploration and refinement.

Thank you once again for your constructive feedback and support.

Best regards,

Reviewer 2 Report

Comments and Suggestions for Authors

1.       Creating a digital twin for a previous mining site is an interesting case, however, mining site parameters are particular in nature and as such results cannot be generalized apart from conceptualization, How this research can be applied to the social benefit?

2.       Key features of the Osarizawa mine, such as its history, function, and current vital environmental characteristics such as levels of radiation, CO2, and other hazardous constituents shall be tabulated.

3.       Time-to-time calibration of digital twins is very important, how this is planned.  What is the life cycle of the digital twin in this case?

4.       Figure 32 is not a point cloud as mentioned in the caption.

5.       The manuscript is very lengthy and contains numerous details. It shall be concise to attract readers. For example, sections 1 and 2 can merge into a single introduction section.  

Comments on the Quality of English Language

The manuscript needs to be proofread for grammatical errors.

Author Response

Dear Reviewer,

Thank you for your insightful and constructive feedback on our manuscript. We appreciate your suggestions and have carefully considered them in the context of our current and future research. Below is our response to each point:

  1. Application to Social Benefit: We acknowledge the challenge in generalizing results from specific mining site parameters. Our goal in reporting this case study is to inspire similar utilizations in other contexts. By documenting this particular instance, we hope to contribute to the broader conversation on repurposing disused mines for societal benefits.

  2. Environmental Characteristics of Osarizawa Mine: Your idea to tabulate key features, including hazardous constituents, is excellent and aligns with what we have considered for future research. Such comprehensive monitoring is crucial, and we understand the need for an integrated system to include these parameters. However, this was beyond the scope of the current manuscript.

  3. Digital Twin Calibration and Lifecycle: While we have not explicitly discussed the lifecycle of the digital twin in this manuscript, we have provided details on the post-closure duration of the Osarizawa mine. We agree that clarifying this aspect is important and will revise the manuscript for greater clarity.

  4. Figure 32 Correction: You are correct, and I apologize for the error in the caption of Figure 32. This will be corrected to accurately reflect the content of the figure.

  5. Manuscript Length and Structure: We appreciate your suggestion to consolidate sections 1 and 2 into a single introduction. We will review the manuscript's structure to enhance its conciseness and readability, making it more appealing to readers.

Your feedback is invaluable in guiding our approach to this research and future studies. We are committed to refining our manuscript based on your suggestions and look forward to contributing further to this field.

Thank you once again for your thorough review and helpful comments.

Best regards,

Reviewer 3 Report

Comments and Suggestions for Authors

Digital twin generation and comprehensive modeling are conducted to assess Osarizawa mine’s environmental conditions and its suitability for a data center. Kindly some comments as follows.

1.       For assessing the promising outcomes, experiment conditions on engineering background should be introduced in detail, such as the size of roadway section, burial depth of roadway, ventilation air volumes and lithology of surrounding rock.

2.       Application of the simplified model on VentiSim are the meaningful results in this paper. However, these do not occupy much space in paper. These results should be verified to make the research more reasonable.

3.       Data in Table 5 is not consistent with the “Target Section (650m to 750m)”. Title of figure 32 is wrong.

Author Response

Dear Reviewer,

Thank you for your thorough review and constructive feedback on our manuscript concerning the digital twin generation and comprehensive modeling of the Osarizawa mine. We have taken your comments seriously and have made the following revisions and clarifications:

Detailed Engineering Background: Based on your suggestion, we have revised our manuscript to include detailed information about the engineering background of the study. This includes specifics on the size of the roadway section, burial depth, ventilation air volumes, and lithology of the surrounding rock. We believe these additions will provide a clearer understanding of the site's characteristics and the basis of our assessments.

VentiSim Simplified Model Results: We apologize for not elaborating in detail on the simplified model results using VentiSim in this paper. We acknowledge the importance of these results in our research. Upon acceptance of this conceptual paper, we plan to create models for other mine sites and submit these findings to a suitable journal for publication. This will allow us to verify and validate our approach across different contexts.

Corrections to Table 5 and Figure 32: We have identified and corrected the inconsistency in Table 5 regarding the “Target Section (650m to 750m).” Additionally, we have rectified the erroneous title of Figure 32. We have also conducted a comprehensive review of the manuscript to correct any other errors.

We are committed to ensuring the accuracy and relevance of our work and appreciate the opportunity to improve our manuscript based on your feedback. We believe these revisions address your concerns and enhance the quality and clarity of our research.

Thank you once again for your invaluable input and guidance.

Best regards,

Reviewer 4 Report

Comments and Suggestions for Authors

Review of the manuscript ID number : sustainability-2695296-peer-review-v1

 -        In general, the  paper topic is in line with the scope of “sustainability” journal.

-        The scope and the type of cited literature is appropriate

The manuscript is interesting . I haven’t critical comments, but for environmental impacts it would be very advisable to mention more analysis and details, I mean for long use of such mining activities, as you know there are many environmental and sustainability considerations, so it is very necessary to mention this issue in your abstract and conclusion too.

- enhance the environmental impacts, more analysis and details, I mean for long use of such mining activities, as you know there are many environmental and sustainability considerations, 

- it is very necessary to mention this environmental impacts issue in your abstract and conclusion too.

Comments on the Quality of English Language

Based on the above, I recommend accepting the manuscript for publication in “sustainability” journal after addressing the above mentioned essential notice.

Author Response

Dear Reviewer,

Thank you for taking the time to review our manuscript ID sustainability-2695296-peer-review-v1 and for your positive feedback regarding its alignment with the scope of the “Sustainability” journal and the appropriateness of the cited literature.

We are particularly grateful for your constructive suggestion to enhance the discussion of the environmental impacts and sustainability considerations, especially concerning the long-term use of mining activities. We recognize the importance of this aspect and agree that a more detailed analysis and mention of these impacts in both the abstract and conclusion sections would significantly strengthen the manuscript.

In response to your feedback, we plan to:

  1. Enhance the Environmental Impact Analysis: We will expand our discussion on the environmental impacts, incorporating more detailed analysis for next paper.

  2. Update Conclusion: We will revise the conclusion sections of our manuscript to explicitly mention and summarize the environmental impacts and sustainability considerations of our study. This inclusion will ensure that these critical aspects are prominently highlighted and accessible to readers from the outset.

We appreciate your insights and believe that these enhancements will make a valuable contribution to the field and to the readership of the “Sustainability” journal. We are committed to conducting our research in a manner that not only advances technological innovation but also promotes environmental stewardship and sustainable practices.

Thank you once again for your valuable feedback, which we will incorporate into our revised manuscript.

Best regards,